# Cognitive reserve against Alzheimer's pathology is linked to brain activity during memory formation

The cognitive reserve (CR) hypothesis posits that individuals can differ in how their brain function is disrupted by pathology associated with aging and neurodegeneration. Here, we test this hypothesis in the continuum from cognitively normal to at-risk stages for Alzheimer's Disease (AD) to AD dementia using longitudinal data from 490 participants of the DELCODE multicentric observational study. Brain function is measured using task fMRI of visual memory encoding. Using a multivariate moderation analysis, we identify a CR-related activity pattern underlying successful memory encoding that moderates the detrimental effect of AD pathological load on cognitive performance. CR is mainly represented by a more pronounced expression of the task-active network encompassing deactivation of the default mode network (DMN) and activation of inferior temporal regions including the fusiform gyrus. We devise personalized fMRI-based CR scores that moderate the impact of AD pathology on cognitive performance and are positively associated with years of education. Furthermore, higher CR scores attenuate the effect of AD pathology on cognitive decline over time. Our findings primarily provide evidence for the maintenance of core cognitive circuits including the DMN as the neural basis of CR. Individual brain activity levels of these areas during memory encoding have prognostic value for future cognitive decline.

Alzheimer's disease (AD) is biologically characterized by the accumulation of amyloid-β (Aβ) plaques, neurofibrillary tangles consisting of aggregated tau protein, and neurodegeneration[1]. Intriguingly, certain individuals resist clinical progression to dementia in their lifetime despite significant AD pathology in their brains[2]. The cognitive reserve (CR) hypothesis addresses this discrepancy. CR is conceptualized as a mismatch between an individual's brain pathology burden and level of cognitive performance due to cognitive and functional brain mechanisms that are not necessarily accompanied by macroscopic structural brain alterations. Recently, a comprehensive whitepaper presented a unified framework for reserve research and defined CR as the "adaptability (i.e., efficiency, capacity, flexibility) of cognitive processes that helps to explain differential susceptibility of cognitive abilities or day-to-day function to brain aging, pathology, or insult"[3].

In alignment with its definition, the most recent research framework operationalizes CR in the form of a moderator variable[4] (see also https://reserveandresilience.com/framework/). As such, it states the requirement of three components for CR research. First, it requires a measure of changes in brain status like brain atrophy or pathology. Second, a quantification of longitudinal changes in cognition theoretically associated with brain status is needed. The third component of this moderation approach is a proposed CR measure, which should moderate the relationship between brain status and cognitive changes. The use of functional neuroimaging methods presents a viable avenue for investigating the neural implementation of CR within this framework. For this purpose, the moderator variable can be represented by the expression of brain activity during cognition using fMRI.

✉ e-mail: Niklas.Vockert@dzne.de; Anne.Maass@dzne.de; Gabriel.Ziegler@dzne.de

Previous studies have only partially been able to address these aspects, even though a wide range of methodologies has been utilized. Most functional neuroimaging studies on CR have identified regions or networks contributing to CR by correlating their expression (activity/connectivity) with a CR proxy like education or IQ instead of investigating their ability to moderate the relationship between aging- or pathology-related brain changes and cognitive performance. For instance, resting-state functional connectivity profiles have been associated with sociobehavioral CR proxies, manifesting at different levels, including ROI-to-ROI[5], global connectivity of a seed region[6,7], global functional connectivity within a network[8] and employing dimensionality reduction to an ROI-to-ROI connectome[9]. Notably, Van Loenhoud et al.[10] recently employed a task-potency method, examining the relationship between whole ROI-to-ROI connectomes in the resting and task state and their association with CR proxies.

Several task-based fMRI investigations pertaining to CR have relied on the Reference Ability Neural Networks Study, wherein participants engaged in 12 cognitive tasks during MRI scanning[10–13] encompassing three tasks each from the four reference abilities of episodic memory, fluid reasoning, perceptual speed, and vocabulary[14]. This comprehensive approach first facilitated the identification of overlapping regions of brain activity across tasks[11–13] and activity patterns exhibiting correlations with IQ[13] and education[11]. Moreover, Stern et al.[15] observed a distinctive spatial pattern of BOLD activity, the expression of which displayed significant correlations with measures of CR as task load increased.

Among the most notable findings, the default mode network (DMN)[9,10] emerged as a potential CR-related region, alongside its individual components such as the left precuneus[13,16], left posterior cingulate[16,17], precuneus and cingulate[15], and medial frontal gyrus[13]. Left prefrontal cortex activity, both within and outside of the frontoparietal network[6], as well as global connectivity of the left frontal cortex[7], were also related to CR. Additionally, there is some evidence for the involvement of the anterior cingulate cortex (ACC) in CR[13,17].

However, most previous attempts neglect that a network underpinning CR should be capable of altering the relationship between aging- or pathology-related brain changes and cognitive performance[3]. Moreover, functional neuroimaging studies on CR are more prevalent in aging research, whereas very few investigations have explored CR in the context of neurodegeneration and AD[7,13]. A major challenge in addressing this gap is to obtain brain activity during cognition in large longitudinal cohorts where AD-related pathological burden is thoroughly quantified.

The primary objective of this study was thus to investigate the neural implementation of CR by identifying task fMRI activity patterns associated with cognitive reserve in a large-scale multicentric cohort of nearly 500 older individuals along the AD spectrum with the use of the moderation framework. Notably, the cohort was enriched in individuals who still perform normally but are at increased risk for developing AD. To accomplish this, we employed a task fMRI paradigm on memory formation to explore CR in the context of episodic memory encoding. Given that episodic memory is among the earliest and most frequently affected cognitive faculties in dementias like AD dementia[18,19], memory-related activity patterns hold particular relevance in CR investigations. As the central hub of episodic memory formation and due to its vulnerability in AD, the hippocampus is further of distinct significance for quantification of AD-related neurodegeneration[20–22]. Our study sought to complement previous approaches by (1) adhering closely to the research framework[4] while (2) identifying a memory-related activity pattern capable of moderating the impact of AD pathology on cognitive performance. Drawing on insights from prior functional neuroimaging studies on CR, we expected that CR-related activity patterns might encompass regions such as the DMN, frontal regions such as the ACC and task-specific regions like the MTL[23]. Our approach takes advantage of a moderation model in a multivariate fashion (utilizing principal component regression) and effectively condensing the multidimensional AD pathological process (reflecting fluid biomarkers and hippocampal atrophy) into a single pathological load (PL) score. We further derived a neuroimaging-based CR score from an individual's expression of the CR-related fMRI activity patterns and show its alignment with educational attainment, a well-established proxy for CR. Finally, we explored the longitudinal implications of this CR index, meticulously examining its potential to modify cognitive trajectories over time.

## Results

### Demographics

Our reserve analysis focused on a sample of 490 older participants of cognitively normal (including first-degree relatives of AD patients and individuals with subjective cognitive decline) and cognitively impaired individuals (with amnestic mild cognitive impairment (aMCI) or Alzheimer's disease dementia (ADD)) who performed task fMRI. Their demographics are presented in Table 1 (mean age: 69.7 ± 5.6 years). We note that of the 68 participants (10.4% of the original sample) that had been excluded from the analyses, the proportion of aMCI (n = 19; 22.9% of the 83 in the original sample) and ADD (n = 12; 36.4% of the 33 in the original sample) was disproportionately higher due to their greater movement during fMRI, response bias, etc. (see Supplementary methods). The sample included slightly more females (53%) than males and was comparably well-educated (14.6 ± 2.9 years of education). The pathological load (PL) reflected biologically defined AD pathology (Aβ, tau, and hippocampal volume) in a single index ranging from 0 to 1 (see "Methods" for details). The sample's mean PL was 0.42 ± 0.3. PL was significantly higher in patients with ADD and aMCI compared to other groups, suggesting its validity with respect to clinical diagnosis.

### Pathological load is associated with cognitive performance

The PL score combines CSF measures of amyloid burden and tau pathology with MRI measures of neurodegeneration into a single score. As a robust marker for disease severity along the AD continuum,

**Table 1 | Demographics of the final fMRI sample**

| | N | Age (years) | Sex (% female) | Education (years) | PACC5 | PL |
|---|---|---|---|---|---|---|
| CN | 152 | 68.89 (5.1) | 63.2 | 14.57 (2.7) | 0.21 (0.5) | 0.33 (0.2) |
| ADR | 51 | 66 (4.7) | 56.9 | 14.49 (2.8) | 0.14 (0.7) | 0.3 (0.2) |
| SCD | 202 | 70.01 (5.9) | 44.6 | 15.2 (2.9) | −0.04 (0.7) | 0.4 (0.3) |
| aMCI | 64 | 72.62 (4.8) | 53.1 | 13.44 (2.8) | −1.22 (0.8) | 0.58 (0.3) |
| ADD | 21 | 73.36 (5.4) | 66.7 | 13.71 (2.8) | −2.97 (1.2) | 0.84 (0.2) |
| All | 490 | 69.73 (5.6) | 53.7 | 14.64 (2.9) | −0.19 (1.0)[a] | 0.42 (0.3)[b] |

Values represent the mean (standard deviation).
Source data are provided as a Source Data file.
*ADD* mild Alzheimer's disease dementia, *ADR* AD patient first-degree relatives, *aMCI* amnestic mild cognitive impairment, *CN* cognitively normal, *SCD* subjective cognitive decline, *PACC5* Preclinical Alzheimer's Cognitive Composite 5, *PL* pathological load.
[a]Nine participants did not have PACC5 scores.
[b]258 participants did not have PL scores due to missing CSF data.

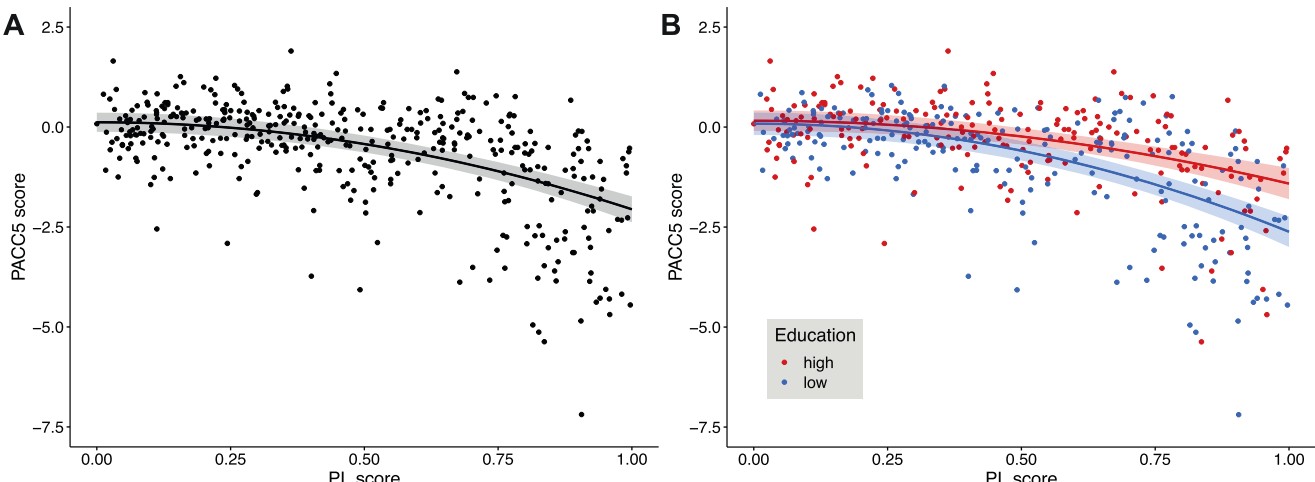

**Fig. 1 | Cognitive performance declines with higher AD pathological load.**
Cognitive performance is represented by the baseline PACC5 score, which was normalized to the unimpaired sample (cognitively normal individuals, subjective cognitive decliners, first-degree relatives of Alzheimer's disease patients).
**A** Quadratic model: PACC5 = $b_0 + b_1 \cdot PL^2 + c \cdot COV$. The black line depicts the predictions of a regression model (with 95% confidence intervals) with a quadratic effect of PL. **B** Same model as in panel (**A**), but with additional terms for years of education and its interaction with the quadratic PL score. Red and blue dots refer to

individuals with high and low education, respectively, as obtained by a median split. Red and blue lines are the predictions of regression models for an individual with average covariate values and 17 (median of the high education group) or 12 years of education (median of the low education group), respectively. Shaded areas refer to the respective 95% confidence intervals. Source data are provided as a Source Data file. COV covariates (see "Methods"), PACC5 Preclinical Alzheimer's Cognitive Composite 5, PL pathological load.

the PL score exhibited substantial associations with cognitive measures derived from neuropsychological testing. Notably, empirical findings indicated a nonlinear relationship (Fig. 1), prompting a comparison of models with linear and quadratic terms for PL. Both models demonstrated a strong link between PL and PACC5, a composite measure of cognitive performance in the memory domain. The model incorporating a quadratic term displayed a superior fit ($p = 5.03 \cdot 10^{-28}$, standardized regression coefficient $\beta = -0.516$ [95% confidence intervals: $-0.602, -0.431$], t(400df) = $-11.864$, $R^2 = 0.385$; Supplementary information provides comprehensive details).

Since this work focuses on reserve as moderation in terms of interactions with PL, we next tested whether education as a well-established CR proxy moderated the impact of PL on cognitive performance ($p = 0.0001$, $\beta = 0.752$ [0.374, 1.129], t(398) = 3.914), which suggested the pivotal role of education in promoting factors that might contribute to the relationship between AD pathology and cognitive abilities (Fig. 1B). Additionally, in the interaction model, a main effect of the (quadratic) PL score was evident ($p = 2.94 \cdot 10^{-4}$, $\beta = -1.223$ [$-1.595, -0.851$], t = $-6.467$), while no further independent main effect of education was observed ($p = 0.575$, $\beta = 0.033$ [$-0.082, 0.147$], t = 0.561). The moderation model demonstrated an $R^2$ value of 0.441, further affirming its predictive capability.

**Identification of a CR-related activity pattern**
We illustrate brain regions exhibiting heightened activity during encoding for subsequently remembered scenes (Fig. 2A, warm colors) or later forgotten ones (cool colors). In exploring cognitive reserve, we aimed to identify those spatial patterns (in terms of local voxel-level weights) from this parametric activity contrast that might moderate the impact of a subject's AD pathological load on cognitive performance using a multivariate moderation approach that predicts performance (see "Methods"). Through cross-validation, we determined that the optimal number of principal components (PCs) for the model was 7, yielding a mean cross-validation $R^2$ of 0.3436 (see Fig. S8 for cross-validation results).

Our investigation unveiled patterns of brain regions contributing both positively (depicted in Fig. 2B, warm colors) or negatively (cool colors) to the moderation of the relationship between AD pathology

and cognitive performance. The former indicates that greater memory-related encoding activity (more positive or less negative) is linked to superior cognitive performance despite the presence of pathology. Conversely, in regions contributing negatively to the moderation patterns, more negative or less positive activity aligns with better cognitive performance amidst increased pathological burden. In other words, individuals with elevated pathology demonstrated better-than-expected cognitive performance when their BOLD signal differences between subsequently remembered and forgotten stimuli were substantial within regions bearing corresponding colors in Fig. 2A, B. These findings highlight the complex interplay between neural activation patterns and cognitive resilience.

To validate and explore the obtained CR-related activity pattern, we then identified clusters with significant contributions (to moderation of the relationship between pathology and cognitive performance) using bootstrapping (Fig. 3 and Table 2). Brain regions with the most positive moderation effects were located bilaterally in the inferior temporal and inferior occipital cortices, including the fusiform gyri and small parts of the right parahippocampal cortex (clusters 1 and 2). To a weaker extent, parts of the frontal cortex also contributed positively to CR, especially bilateral inferior frontal gyri, including opercular, triangular and orbital parts (clusters 3 and 4) as well as parts of right PFC (cluster 8 in Table S3). The strongest negative contributions to moderation were observed in bilateral precuneus, cuneus, posterior cingulate cortex (PCC; cluster 5). Slightly weaker negative coefficients were found in the bilateral inferior parietal cortex around the angular gyrus (clusters 6 and 7).

Interestingly, CR-related activity patterns did not predominantly reflect regions showing atrophy in the DELCODE cohort (mostly found in the hippocampus and medial temporal lobe, Fig. 2C), with minor overlaps in left hippocampus, precuneus and PCC. Overall, this lack of overlap between identified CR-related regions and regions of strongest atrophy was supported by a low correlation of $-0.102$ (Fig. 2B, C).

To delve deeper into our comprehension of the identified pattern, we conducted an examination of the overlap between the CR-related activity pattern and the generic subsequent memory activity pattern (illustrated in Fig. 2A). A substantial concurrence between these patterns was observed in the most extensive clusters of CR-related

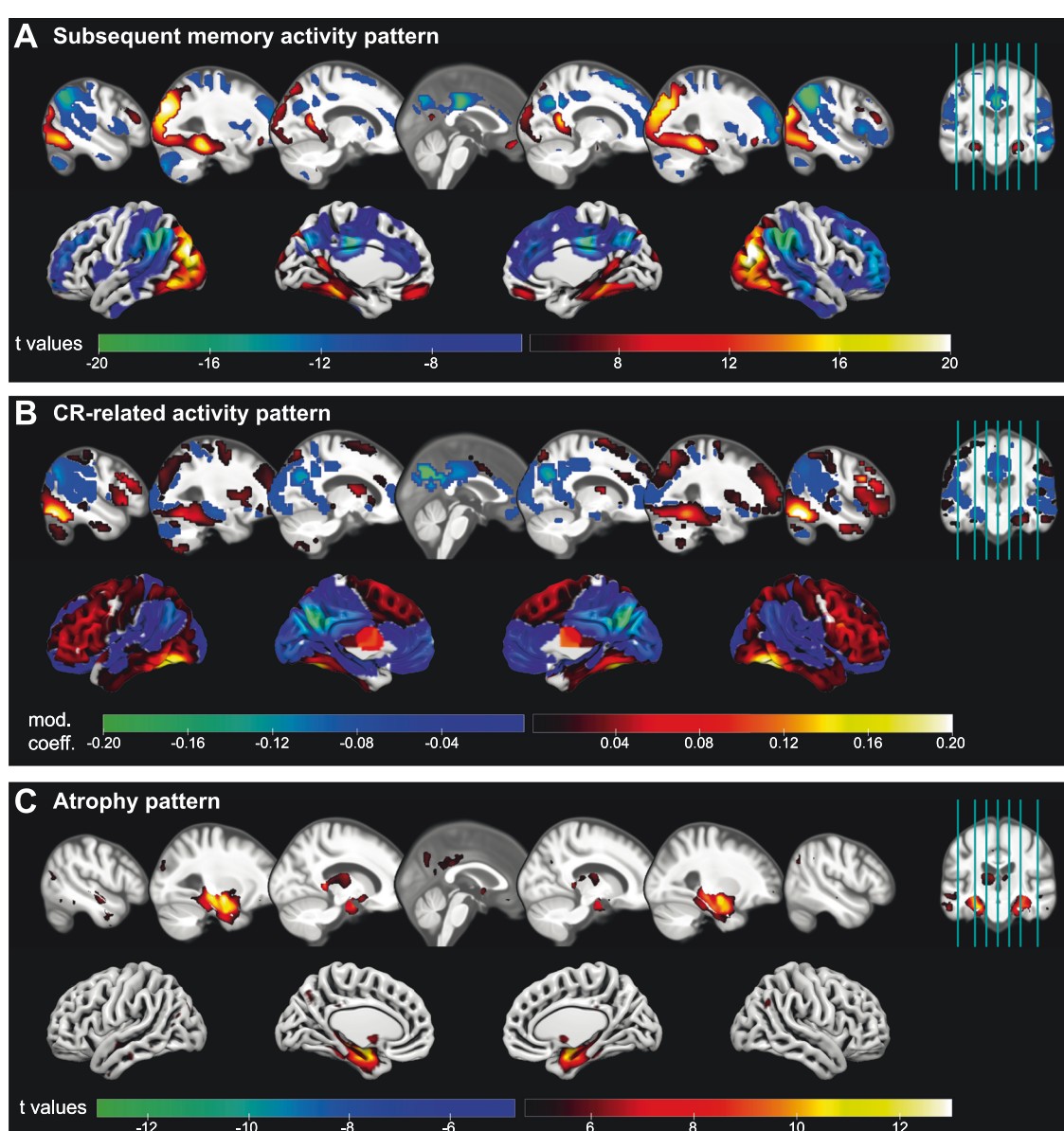

**Fig. 2 | CR-related activity pattern that moderates effects of pathology.**
**A** Activation (hot colors) and deactivation (cool colors) during encoding of subsequently remembered compared to subsequently forgotten stimuli as identified by t-contrasts of the subsequent memory regressor in the whole fMRI sample. T values of voxels with $p_{FWE} < 0.05$ are shown. **B** Group-level CR-related activity pattern that when expressed minimizes effects of AD pathology on cognitive performance as identified via a multivariate approach. The net contribution (moderation coefficient; positive/hot and negative/cool colors) of every voxel to the CR pattern is displayed (unthresholded). **C** Atrophy pattern in the whole baseline DELCODE sample as obtained by a VBM GM analysis of CN participants vs ADD patients. T values of voxels with $p_{FWE} < 0.05$ are shown. Source data are provided as a Source Data file. ADD Alzheimer's disease dementia, CN cognitively normal, CR cognitive reserve, GM gray matter, VBM voxel-based morphometry.

deactivations, notably in regions such as the precuneus, posterior cingulate cortex (PCC), angular gyrus and ACC, and activations, particularly in inferior temporal areas (Fig. 4, blue). Within those regions, a higher reserve is reflected by an increase in the amplitude of the task-related activation/deactivation. However, it was also apparent that reserve is not uniformly contributing across this task-active network and that certain regions exhibiting significant (de)activation during successful memory encoding did not substantially contribute to cognitive reserve at all (such as portions of the parietal, frontal, temporal, particularly occipital cortex, as well as the cerebellum and basal ganglia).

Moreover, a striking observation emerged in a few brain regions where the valence of the coefficients did not align, a phenomenon we term "discordant" (Table 2, Fig. 4). For instance, voxels surrounding the calcarine sulci (parts of the bigger cluster 5), bordering the cuneus and precuneus, displayed activation during successful encoding but exhibited a negative contribution to CR (CR-SM+, green in Fig. 4). This trend was also observed in the left hippocampus and medial orbitofrontal regions. Conversely, positive contributions to CR were evident in certain right frontal areas, such as the insula and mid/superior orbitofrontal cortex (e.g., parts of clusters 3 and 4; CR+SM-, yellow in Fig. 4), despite subsequent memory-related deactivation. Taken together, the correlation between the voxelwise CR coefficients and SM contrast values was found to be 0.384. This suggests that predominantly showing activation patterns closer to the typical activation/deactivation might

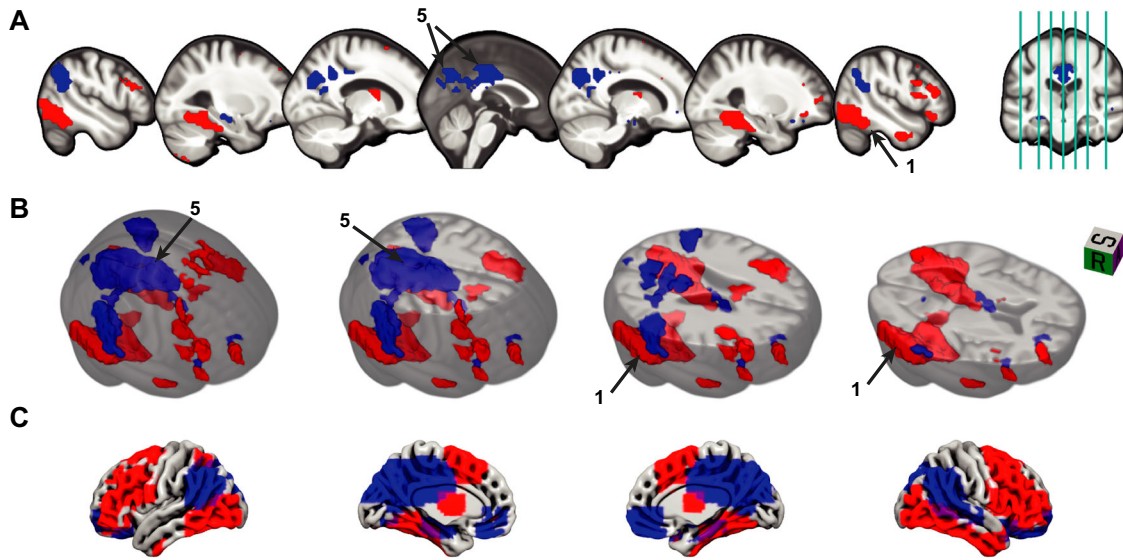

**Fig. 3 | Significant regions in the CR-related activity pattern.** Several clusters of voxels were determined via bootstrapping whose contribution to the CR pattern ($w_i$) was found to be significant ($p < 0.05$, see "Methods"), displayed as **A** mosaic (multislice) view, **B** 3D view and **C** surface view. Displayed numbers refer to the clusters described in Table 2 with peaks in the following brain structures. 1: right inferior temporal cortex, 5: left precuneus. In panel (**B**), small clusters have been removed for illustrative purposes. It is important to note that the CR pattern is multivariate in nature, interpretable as a whole and cluster descriptives are reported for transparency of obtained non-negligible coefficients contributing to the pattern. Source data are provided as a Source Data file. CR cognitive reserve.

**Table 2 | Significant clusters in CR-related activity pattern**

| # | Mean $w_i$ | Size (voxels) | % concordant | Peak (x, y, z) | Peak structure |
|---|---|---|---|---|---|
| 1 | 0.100 | 623 | 100 | 49, −63, −17 | Temporal_Inf_R |
| 2 | 0.085 | 594 | 100 | −49, −66, −17 | Occipital_Inf_L |
| 3 | 0.055 | 149 | 79.19 | 49, 14, 28 | Frontal_Inf_Oper_R |
| 4 | 0.075 | 119 | 93.28 | −49, 25, 24 | Frontal_Inf_Tri_L |
| 5 | −0.105 | 873 | 90.38 | 0, −60, 32 | Precuneus_L |
| 6 | −0.067 | 235 | 97.45 | −45, −63, 35 | Angular_L |
| 7 | −0.048 | 213 | 100 | 52, −56, 39 | Parietal_Inf_R |

Structures and peak voxels were identified in MRIcroGL, using the AAL (Automated Anatomical Labeling) atlas[76]. $w_i$ refers to the CR coefficient of a voxel $i$. % concordant refers to the proportion of voxels in the cluster that have the same valence (sign) for the CR coefficient and the parametric subsequent memory contrast coefficient as shown in Fig. 4. A concordant region is one where a higher (lower) activity reduces effects of pathology and which is typically activated (deactivated) during the task. Clusters smaller than 50 voxels (voxel size: 3.5 × 3.5 × 3.5 mm³) have been omitted. It is important to note that the CR pattern is multivariate in nature, interpretable as a whole and cluster descriptives are reported for transparency of obtained non-negligible coefficients contributing to the pattern. Source data are provided as a Source Data file.
CR cognitive reserve.

support cognitive functioning. On the other hand, more complex region-specific multifaceted relationships between these neural signatures and cognitive reserve might exist, indicated e.g., by discordant voxels.

Next, we exemplify how subsequent memory-related activity moderates the detrimental effect of pathology (PL score) on cognitive performance in two brain regions located in the right inferior temporal cortex (Fig. 5A) and around bilateral cuneus/precuneus/PCC (Fig. 5B), respectively (taking clusters 1 and 5 from Fig. 3 and Table 2). The moderation effect has unveiled a notable phenomenon: as levels of pathological load (PL) rise, the disparities in cognitive performance between individuals with high and low levels of (de)activation become increasingly apparent. Among individuals with high PL, those with high SM contrast values have cognitive ability at the level of individuals with low PL.

### CR score moderates effects of pathology on cognitive performance, also longitudinally

Utilizing the CR-related activity pattern that we identified above, we next derived individualized CR scores. To ascertain its validity as an

indicator of cognitive reserve, we expected it to (1) moderate the effect that pathology has on independent cognitive performance measures, (2) moderate longitudinal cognitive decline, and (3) be correlated with sociobehavioral proxies of CR according to the consensus research criteria[3].

Our results affirm the first criterion, demonstrating a moderation effect of the CR score on the relationship between the (quadratic) PL score and cognitive performance across various cognitive tests (Fig. 6A). This moderation effect was evident for the latent memory factor ($p = 8.38 \cdot 10^{-12}$, $\beta = 0.381$ [0.277, 0.485], t(218) = 7.224), the domain-general factor ($p = 2.15 \cdot 10^{-8}$, $\beta = 0.325$ [0.215, 0.435], t(218) = 5.814) and the PACC5 score ($p = 9.15 \cdot 10^{-15}$, $\beta = 0.447$ [0.341, 0.552], t(214) = 8.338), which was originally used in identifying the CR-related activity pattern. Importantly, this moderating effect was not only observed in individuals with cognitive impairment, i.e., aMCI and AD patients but also when analyzing the same models only in cognitively unimpaired individuals (memory factor: $p = 3.91 \cdot 10^{-5}$, $\beta = 0.273$ [0.145, 0.400], t(171) = 4.223; domain-general factor: $p = 0.0010$, $\beta = 0.220$ [0.091, 0.349], t(171) = 3.358; PACC5: $p = 3.77 \cdot 10^{-6}$, $\beta = 0.301$ [0.177, 0.426], t(170) = 4.781). This emphasizes that the fMRI activity patterns

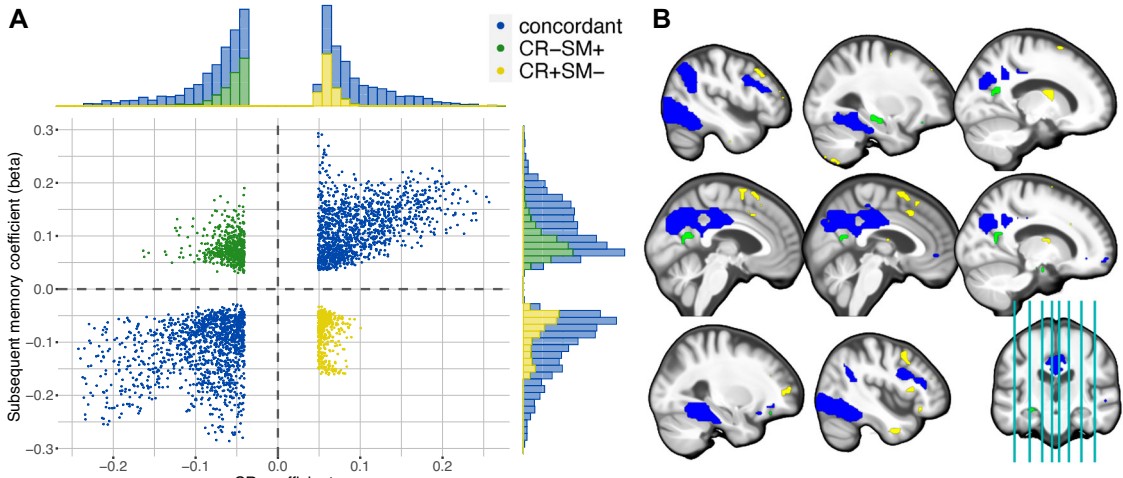

**Fig. 4 | CR pattern and the subsequent memory contrast predominantly align.**
**A** The scatter plot displays the CR coefficients $w_i$ and subsequent memory contrast coefficients (beta) for every voxel with significant contribution to CR. They form three groups: 1. A concordant where both coefficients have the same sign (blue); 2. positive CR coefficient, but negative subsequent memory beta (CR+SM-; yellow); 3. negative CR coefficient, but positive subsequent memory beta (CR-SM+; green). The histograms display the frequency of the voxels in the corresponding groups.

The gray dashed lines separate the four quadrants. **B** The CR-related activation pattern is shown color-coded corresponding to the colors in panel (**A**). It is important to note that the CR pattern is multivariate in nature, interpretable as a whole and cluster descriptives are reported for transparency of obtained non-negligible coefficients contributing to the pattern. Source data are provided as a Source Data file. CR cognitive reserve, SM subsequent memory.

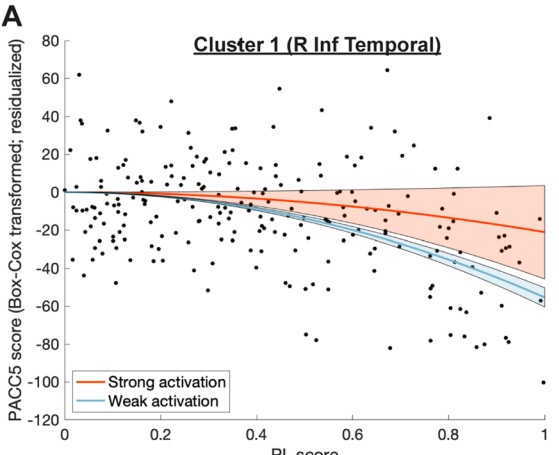
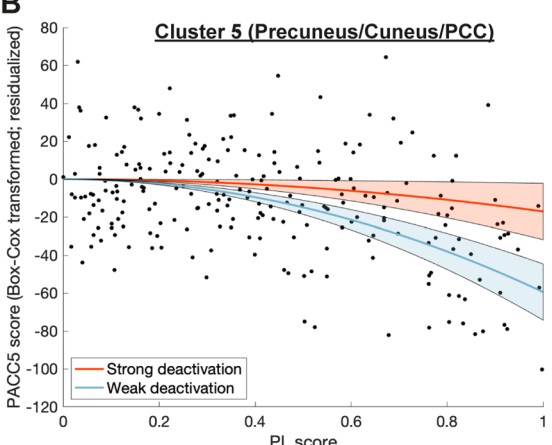

**Fig. 5 | Subsequent memory-related activity moderates the relationship between PL and PACC5.** The relationship between the PL score and the PACC5 score (Box-Cox transformed and residualized for covariates) is moderated depending on the subsequent memory-related activity in two previously identified clusters (see Table 2 or Fig. 3). **A** Moderation effect of activation in cluster 1 located around the inferior temporal cortex including fusiform gyrus (positive moderation coefficients). **B** Moderation effect of deactivation in cluster 5 including bilateral cuneus and precuneus as well as posterior cingulate (negative moderation

coefficients). The red lines in both panels depict the predicted PACC5 score for a hypothetical individual with an activation 1 SD above the mean, the blue lines for an activation 1 SD below the mean in the respective cluster. The shaded areas represent the 95% confidence intervals. Black dots represent the individual subjects' values for PL and (transformed + residualized) PACC5. Source data are provided as a Source Data file. PACC5 Preclinical Alzheimer's Cognitive Composite 5, PL pathological load.

associated with CR might benefit a broad spectrum of cognitive abilities.

Furthermore, the same CR score exhibited a weaker yet significant moderation effect on the association between AD pathology and cognitive performance (see Fig. 6B) for the latent memory factor ($p = 1.17 \cdot 10^{-6}$, $\beta = 0.202$ [0.094, 0.310], t(244) = 3.697), domain-general factor ($p = 0.020$, $\beta = 0.138$ [0.022, 0.254], t(244) = 2.346) and PACC5 score ($p = 1.35 \cdot 10^{-6}$, $\beta = 0.271$ [0.163, 0.378], t(240) = 4.958) in the remaining sample without a PL score (due to missing CSF data). Here, the PL score was replaced by hippocampal atrophy (squared, as the PL score).

In a longitudinal context, lower hippocampal volumes at baseline worsened cognitive decline rates in the PACC5 score ($p = 5.24 \cdot 10^{-8}$, $\beta = -0.140$ [−0.191,−0.092], t(158.6) = −5.717) in the whole sample. The CR score attenuated this relationship (three-way interaction of CR score, atrophy and time; $p = 1.19 \cdot 10^{-4}$, $\beta = 0.118$ [0.060, 0.179], t(124.5) = 3.974), suggesting that activity patterns during memory encoding hold the potential to influence cognitive trajectories in the face of pathology (Fig. 6C, D). There was also an interaction of the CR score with hippocampal atrophy ($p = 3.28 \cdot 10^{-11}$, $\beta = 0.338$ [0.240, 0.436], t(487.6) = 6.790) as in the cross-sectional models. Results for these analyses when using the PL score instead of hippocampal atrophy in the CSF-

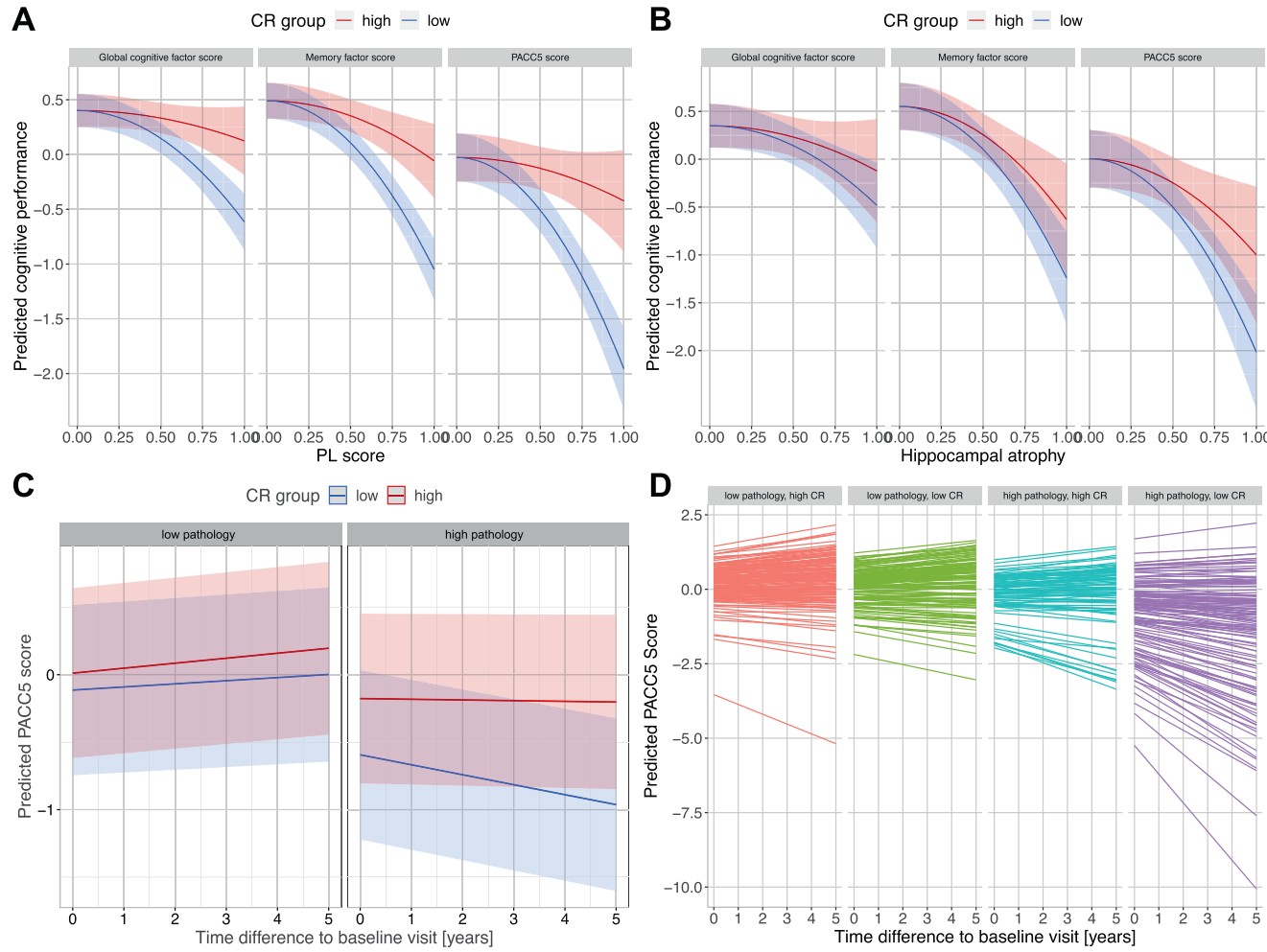

**Fig. 6 | CR score is linked to cognitive performance cross-sectionally and longitudinally. A** The relationship between the PL score and cognitive performance at baseline is moderated by the CR score. Cognitive performance is represented by three different scores: a global cognitive factor score, a memory factor score and the PACC5 score (previously used for identification of the CR-related activity pattern). Cognitive performance was predicted using the respective regression model for an average individual with high (above median; red curve) or low (below median; blue curve) CR score. **B** In the sample without PL score, the CR score moderates the relationship between hippocampal atrophy and cognitive performance. Prediction procedure equivalent to the one in panel (**A**), but based on a model that was fit in the sample without PL scores and thus contains hippocampal atrophy instead of the PL score as an independent variable. **C** The pathology-dependent differences in longitudinal trajectories of cognitive performance are

ameliorated by the baseline CR score. The PACC5 scores at a 5-year follow-up were predicted using the previously described LME (see "Methods" section) fitted on the original longitudinal data for an average individual with high (above median; red line) or low (below median; blue line) CR score and high or low pathology (here represented by high and low hippocampal atrophy corresponding to the 25th and 75th percentile). **D** Predictions of all individual cognitive trajectories. Individuals were categorized as high/low CR based on their above/below average CR scores and as low/high pathology based on their below/above average hippocampal atrophy. Shaded areas in panels (**A**–**C**) denote 95% confidence intervals. Source data are provided as a Source Data file. CR cognitive reserve, LME linear mixed-effects model, PACC5 Preclinical Alzheimer's Cognitive Composite 5, PL pathological load.

subsample are presented in the Supplementary information. Moreover, the CR score exhibited a positive correlation with education across the entire sample ($p = 0.012$, r(487) = 0.114 [0.025, 0.201]). Collectively, these findings robustly support the contention that the obtained CR score is intimately associated with cognitive reserve, both in cross-sectional and longitudinal assessments.

## Discussion

In this study, we combined multiple ideas to investigate the neural implementation of cognitive reserve utilizing task fMRI data from a substantial sample comprising 490 participants. First, we employed the most contemporary research criteria governing CR assessment via functional neuroimaging[3,4]. Second, in order to enable this moderation approach, we reduced the dimensionality of AD biomarkers in a non-linear fashion, introducing a novel data-driven index quantifying Alzheimer's disease-related pathological load. Third, we pioneered a

novel multivariate approach to modeling reserve, which uncovered a task-related functional activity pattern capable of moderating the impact of brain pathology on cognitive performance. Fourth, we provided both cross-sectional and longitudinal validation of the proposed activity pattern of cognitive reserve. Our findings illuminate a compelling connection: older individuals whose brain responses during successful memory encoding more closely align with this identified pattern exhibit diminished cognitive deficits when faced with AD pathology. Moreover, a more pronounced expression of this activity pattern was associated with a slower rate of cognitive decline over longitudinal follow-ups and especially attenuated the detrimental effect of pathology on cognitive trajectories. On top of that, the activity patterns also reflected the amount of AD pathology to a certain degree, linking it to cognitive performance in an intricate relationship.

In healthy young individuals and older adults, episodic memory encoding is associated with a highly replicated canonical pattern of

brain activation in some regions and deactivation in other regions[24,25]. We found that a more pronounced expression of this canonical activation/deactivation pattern was associated with higher cognitive reserve. CR was especially characterized by a stronger activation during memory encoding in inferior occipital and inferior temporal areas including the fusiform gyrus, i.e., parts of the ventral visual stream. Some frontal areas showed a similar contribution to CR, though to a smaller extent. CR was further characterized by stronger deactivation in the posterior cingulate cortex, precuneus, cuneus and lateral parietal cortex including angular gyrus, regions that are considered to be part of the DMN[26]. This combined pattern of inferior temporal activations and DMN deactivations has previously been associated with better memory performance in older adults[25,27,28] and, more recently, with severity across the Alzheimer's risk spectrum[29].

These findings shed new light on the neural implementation of CR. As the majority of brain regions showed concordant activity for CR and successful memory encoding, cognitive reserve primarily seems to be associated with continued recruitment of core cognitive circuits. This indicates that some individuals are able to maintain functional integrity in parts of the core cognitive circuitry despite the presence of AD pathology. Generally, significant decreases in fMRI activity in regions of the DMN have been shown to co-occur with amyloid deposition in older adults[30,31]. Furthermore, AD has been characterized by impairment of regional cerebral blood flow and regional glucose metabolism during resting-state, predominantly in temporo-parietal regions[32,33]. Hence, the ability to maintain core functional circuits might represent resilience against pathological changes like neurodegeneration and Aβ accumulation, possibly accompanied by conservation of glucose metabolism. Our findings suggest that this ability is central to CR. The neural mechanisms underlying this ability are still unclear. An attempt to identify additional factors related to the CR score (see Supplementary information and Fig. S2) did not reveal associations with resting-state functional connectivity within the DMN or other large-scale networks nor mean task signal, although a partial moderation of the DMN BOLD signal by resting-state fluctuation amplitude cannot be excluded[34]. One possibility is that pathology within core circuitry can be counteracted by non-affected neural populations. The weak yet significant relationship between the CR score and GM volumes in CR-related regions indeed suggests it has a small contribution (Fig. S2). This possibility is further supported by the pattern of spread of tau pathology within brain tissue where selective cellular vulnerability leads to dysfunction in specific neuronal subpopulations[35,36]. Another possibility is that there are individual differences in pathology that were not quantified in the PL score. These include inflammation, vascular supply and clearance[37–40]. It is possible that individuals who are capable of maintaining function in core circuitry despite of tau, amyloid and hippocampal neurodegeneration have less expression of these additional pathologies. A weak negative correlation between white matter hyperintensity volumes and the CR score supports this hypothesis (see also Fig. S2). In conclusion, our findings primarily support the notion that some CR factors might operate within core circuits themselves above compensatory activity discordant with successful encoding activity.

Besides the moderation effect, the CR score, which represents CR-related activity during successful memory encoding, was also weakly correlated with different measures of pathology and mediated the effect of PL on PACC5 scores (see Supplementary information). These findings point to a more intricate relationship between pathology, brain activity and cognitive performance, where low CR/hyperactivity and pathology could promote each other in a vicious cycle[30,41–45].

Some brain regions within the canonical episodic memory activity network were not associated with CR. Visual areas showed strong activation in the subsequent memory contrast due to the visual nature of the memory task but did not contribute substantially to cognitive reserve. These regions have not been discussed much in the context of

CR, although there has been scattered evidence for a contribution of inferior and middle occipital regions[12].

Encoding-related activity in the hippocampus was discordant with cognitive reserve. Thus, although the hippocampus is well-known to be activated during successful memory encoding, weaker left hippocampal activity during encoding was associated with better cognitive performance in the presence of AD pathology. This aligns with observations regarding hyperactivity of the hippocampus in an Aβ- and especially tau-dependent manner that is not related to better cognitive performance[30,46,47]. An absence of pathology-related overactivation in the hippocampus might actually be beneficial for cognitive performance and clinical progression. The calcarine sulci (part of cluster 5) and right medial orbitofrontal regions show the same kind of discordant activity for successful memory encoding versus CR. Some frontal regions, including the insula and mid/superior orbitofrontal cortex, were deactivated during successful memory encoding and have positive moderation coefficients, indicating better cognitive performance with weaker deactivation. Yet, it is also conceivable that decreased activity in these regions itself is not actually beneficial for cognitive performance but systematically co-occurs with beneficial activity changes in other regions. Similar to our findings regarding discordant regions, previous reports have identified comparable phenomena. For instance, Elman et al.[48] discovered a cluster in the medial parietal cortex in which deactivation was parametrically modulated by the level of memory encoding detail across the entire sample of young and older adults with and without Aβ. In this cluster, greater deactivation was associated with higher detail recall. However, young adults exhibited no such deactivation, resembling our observations of discordant regions.

Generally, the strongest negative contributions to CR were observed in the cuneus, angular gyri, PCC and particularly the precuneus. This is in line with a large body of evidence highlighting the role of the DMN in cognitive reserve. For instance, deactivations of the left precuneus[13,16] and posterior cingulate[16] were associated with CR in previous studies. Moreover, the precuneus, together with the cingulate gyrus, contributed negatively to some aspects of a CR-related fMRI pattern[15]. Connectivity-based methods provide additional evidence for DMN contribution. For instance, using a task-potency method, which captures a brain region's functional connectivity during task performance after adjusting for its resting-state baseline, the DMN has been found to be the predominant contributor to a task-invariant CR network[10]. Furthermore, inhibitory information flow from the inferior temporal cortex, which showed a positive contribution to CR in this study, to the precuneus has been associated with better memory performance in an independent cohort of older adults using the same encoding task[49]. Additionally, Stern et al.[9] suggested that connections involving the DMN might be weaker at rest in individuals with higher IQ. Our findings provide further evidence that stronger deactivation of some DMN regions is related to CR.

The ACC has also previously been identified in the context of cognitive reserve, e.g., as part of the task-invariant CR pattern of Stern et al.[13]. Moreover, greater volume and metabolism in the ACC were found to be related to higher levels of education[5]. It was further identified as part of a "resilience signature" whose metabolism was associated with global cognitive performance in cognitively stable individuals over 80 years[50]. Here, we only find a few significant voxels around the ACC with negative contributions to CR, providing weak additional evidence for the involvement of the ACC in cognitive reserve.

The strongest positive contributions to CR were observed in the fusiform gyri and surrounding temporal to inferior occipital regions. With respect to the fusiform gyrus, there has been both evidence for negative as well as positive contributions to CR[12,15]. Further, some frontal regions have been proposed to play a role in cognitive reserve. For instance, Franzmeier et al.[7] discovered that global connectivity of

the left frontal cortex attenuated the relationship of precuneus FDG-PET hypometabolism (as proxy for AD severity) on lower memory performance in amyloid-positive individuals with aMCI. The left frontal cortex also showed positive contributions to CR in our study (cluster 4 in Table 2; see also Fig. 3). Likewise, left prefrontal cortex connectivity both within and outside the frontoparietal network has been found to correlate with fluid intelligence as a proxy of CR[6].

The expression of this CR activity pattern in an individual, as represented by the task-derived CR score, further fulfills the latest research criteria on CR. First, the CR score moderates the effect of pathological load on cognitive performance. Hence, individuals with lower cognitive reserve scores show a stronger nonlinear decline in their cognitive abilities with increasing pathological burden compared to individuals with higher levels of cognitive reserve. Analogously, individuals with higher pathological burden benefit more from higher levels of CR compared to the ones with low pathology. The CR score retains its disease-moderating characteristic in the context of multiple different cognitive scores like an independent composite memory measure as well as a very broad measure of cognitive abilities spanning learning and memory, language, visuospatial abilities, executive function and working memory. This reveals a certain robustness of the moderating effect of the CR score and supports its validity. Importantly, the CR score ameliorated the negative influence that baseline pathology had on cognitive trajectories over longitudinal follow-ups, stressing its significance not only for present cognitive abilities but also for their development over time. Furthermore, the CR score was related to education, even though the correlation was found to be rather low to moderate. On the one hand, this could mean that our CR score might capture cognitive reserve incompletely due to the apparent task dependency. On the other hand, the correlation should not be close to 1 either, since education itself is only a proxy of CR. Thus, education and CR, as identified via functional neuroimaging approaches, do share parts of their variance but are also partially independent.

Taken together, the moderating effect of the obtained CR score and its relation to another sociobehavioral CR proxy suggest it as a valid, even though incomplete representation of overall cognitive reserve. It also provides evidence that the underlying network indeed contributes to CR, at least in context of the incidental encoding task at hand.

This study has a number of limitations. The approach was enabled by dimensionality reduction of ATN via the t-SNE method, which provided us with a useful tool for quantifying pathological load. Yet, the PL score is a purely cross-sectional construct that is agnostic for the order of events along the disease progression towards Alzheimer's disease, and it may be an oversimplification to represent the ATN system of AD biomarkers by a single variable. Likewise, while hippocampal atrophy is a key feature of AD, it is an oversimplification to represent neurodegeneration solely by hippocampal measures. Strong associations with the three biomarkers (see Supplementary information) as well as with cognitive performance nevertheless suggest the PL score as a meaningful index of overall disease severity (rather than disease progression) for the purpose of this study. In fact, usage of three alternative methods (PCA, spectral embedding, diffusion pseudotime[51]) produced highly correlated PL scores that in turn yielded very similar CR coefficients (see Figs. S13, S14), indicating that, within reason, the method for obtaining a PL score has only a minor influence on the results. Regional quantification of tau pathology using tau-PET, rather than the regionally agnostic fluid biomarker-based tau measurement here, could, in future, help to determine to what extent the recruitment of the CR regions depends on the absence of tau pathology in those regions. Despite its limitations, t-SNE or other nonlinear dimensionality reduction tools like spectral embedding and Laplacian Eigenmaps among many others (see e.g., Van Der Maaten et al.[52] for an overview) or even trajectory inference methods like diffusion pseudotime (see e.g., Saelens et al.[53]) might be useful in many other studies investigating multidimensional disease-related phenomena. Furthermore, our multivariate regression approach relied on reducing the complexity of fMRI data via PCA. As a consequence, the moderation analysis might represent an incomplete characterization of CR in the context of successful memory encoding. As the difference in mean cross-validation $R^2$ was small in comparison to its variability, other choices for the number of principal components (from 2 to 9) also appear reasonable. Consequently, we would like to note that this model hyperparameter is a non-negligible determinant of the results, susceptible to the bias-variance tradeoff. In particular, a model with only two PCs was found to be too restrictive considering the obtained worse model generalization (in cross-validation) and further validation analysis results (see Supplementary results and Fig. S11). To support the confidence in the presented results, we provide an illustration of the overlap of CR regions in dependence of the number of PCs and a comparison of models with different numbers of PCs in Fig. S9.

Of note, an ad hoc definition of a similarity score representing an individual's activity similarity to group-level successful encoding activity in cognitively normal individuals (similar to the idea of the FADE[54] or FADE-SAME score[25]) also shows considerable moderation effects of pathology on cognitive performance in our sample (see Supplementary information). Yet, our proposed approach offers some advantages. First, it explicitly assumes and tests the moderation as specified in the recent consensus framework on the level of brain activity patterns using task fMRI. Second, this multivariate moderation approach allows for more specific findings, such as the identification of above presented discordant regions. Third, the moderation approach can be extended to other indicators of pathology, memory-independent contrasts, multi-task data or other forms of reserve-related imaging features such as resting-state fMRI.

The approach was applied under the simple working assumption that pathology is the initial driver of cognitive decline and that individual variations of task-related activity potentially affect this relationship. However, since the proposed main approach using cross-sectional data exploits simple correlations and symmetric interaction terms, it does allow for several alternative causal interpretations with inter-changed roles of key variables. Future longitudinal studies might focus deeper on the empirical plausibility of these alternative patterns of interplay.

Moreover, one might consider learning the moderating function of brain activity f(A) and its interactions with pathology (see "Methods" Eq. (1)) more directly using neuronal nets and other data-driven approaches[55]. Additionally, CR is treated as a static measure in this study, contrasting with CR's conceptualization as a dynamic entity, susceptible to variation over time[56,57]. However, our approach theoretically allows to represent CR in a dynamic manner by incorporating longitudinal fMRI data. Furthermore, the current study focuses on an approach that assumes sample-level identification of reserve patterns, while ignoring the possibility that CR might actually be implemented differently across different subpopulations or disease stages. For instance, CR-related activity patterns might differ between early and late disease stages[58] or males and females (see Figs. S17, S18). Our search (or model input-) space was limited to the widespread successful encoding network, restricting our ability to identify regions that show compensatory activity to these areas. An analysis extending this search space to all gray matter indicated CR regions essentially as a resemblance of the successful encoding network (Fig. S10). However, we also advise against overinterpretation of these findings since we cannot exclude the possibility of compensatory activity in other tasks or fMRI contrasts. Last, we remark on the dependency of the CR-related activity pattern on the task and contrast at hand. We recognize the efforts of task-invariant approaches to identify an underlying pattern of CR that is task-independent. Nonetheless, apart from task-specific components, the presented CR activity pattern most likely also

contains task-invariant components, e.g., the DMN and ACC. Furthermore, since our contrast probes memory, which is the earliest and most strongly affected faculty in AD, our specific CR-related activity pattern is of great significance in the context of (AD) dementia.

In summary, using a multivariate approach to modeling CR, we have identified a memory encoding-based activity pattern of cognitive reserve in the context of successful memory encoding according to the latest research definitions. We provide further evidence for the hypothesis of a generic role of the DMN and potentially ACC in cognitive reserve. Additionally, we identified regions less commonly associated with cognitive reserve like the fusiform gyrus and some frontal regions. Overall, our findings suggest an enhanced maintenance of core cognitive circuits as the primary neural implementation of cognitive reserve. Consequently, interventional efforts should incorporate methods to maintain the functionality of core cognitive circuitry, for instance, through direct brain stimulation, in order to ameliorate future cognitive decline. However, adequate judgment about compensation in the context of cognitive reserve should be based on further studies specifically designed for its investigation, involving multi-task and -contrast information as well as manipulation of task demand.

Ultimately, more longitudinal studies are necessary to assess the degree of dynamics of CR over time and its ability to modulate trajectories of cognitive decline. Furthermore, CR patterns have only been assessed on the group level, assuming CR works similarly across all individuals. Upcoming approaches should account for individual differences in functional (re)organization, considering individual-level expressions of cognitive reserve.

## Methods

### Sample

The sample is part of the DZNE-Longitudinal Cognitive Impairment and Dementia Study (DELCODE) study, a multicentric observational study of the German Center for Neurodegenerative Diseases (DZNE). It focuses on the characterization of subjective cognitive decline (SCD) in patients recruited from memory clinics, but additionally enrolled individuals with amnestic mild cognitive impairment (aMCI), mild AD dementia patients, AD patient first-degree relatives (ADR), and cognitively normal (CN) control subjects. Participants were scheduled for annual follow-up appointments over five years. More detailed information about the study has been provided previously[59]. Our analyses were based solely on the baseline measures of the participants, with the exception of annually acquired cognitive data, which was used to assess cognitive trajectories and how they might be modified depending on an individual's cognitive reserve. The whole baseline sample comprised 1079 participants, of which 558 participants had undergone an MRI session including the fMRI task. 442 participants had cerebrospinal fluid (CSF) data available and could thus be used for creating the PL score. The final fMRI sample used in the subsequent CR analysis consisted of 490 participants after quality control and outlier exclusion, of which 232 had CSF measures and thus a PL score. Of the 490 participants, 152 were CN, 202 had SCD, 64 aMCI and 21 had a clinical diagnosis of AD dementia. The sample also contained 51 first-degree relatives of AD patients.

CN was defined as having memory test performances within 1.5 SD of the age-, sex-, and education-adjusted normal performance on all subtests of the CERAD (Consortium to Establish a Registry of AD test battery). ADR had to achieve unimpaired cognitive performance according to the same criteria. SCD was defined as the presence of subjective cognitive decline as expressed to the physician of the memory center and normal cognitive performance as assessed with the CERAD[60]. Participants were classified as aMCI when displaying an age-, sex-, and education-adjusted performance below −1.5 SD on the delayed recall trial of the CERAD word-list episodic memory tests. aMCI patients were non-demented and had no impairment in daily

functioning. Finally, only participants with a clinical diagnosis of mild AD dementia[61] obtaining ≥18 points on the Mini Mental State Examination (MMSE) were included in DELCODE. All participants were 60 years or older, fluent speakers of German and had a relative who completed informant questionnaires. Exclusion criteria are described in Jessen et al.[59].

The study protocol was approved by the Institutional Review Boards of all participating study centers of the DZNE[59]. The process was led and coordinated by the ethical committee of the medical faculty of the University of Bonn (trial registration number 117/13). All relevant ethical regulations were complied with. All participants provided written informed consent. DELCODE has been registered with the DRKS (accession number: DRKS00007966).

Disaggregated sex and gender data have not been collected. They were not considered in the study design. Sex was self-reported, and the distributions were 263 females and 227 males in the analyzed sample. Sex and gender-based analyses were not performed, as the overarching goal of the study was to determine a general fMRI-based pattern of cognitive reserve. A brief additional disaggregated analysis for males and females on cognitive reserve is presented in the Supplementary information.

### Cognitive tests

An extensive list of all neuropsychological tests administered in DELCODE is provided elsewhere[59]. In our analysis we use composite scores from those tests, namely the Preclinical Alzheimer's Cognitive Composite 5 (PACC5)[62], a neuropsychological composite measure designed to index cognitive changes in the early phase of AD, and a latent memory factor derived from a confirmatory factor analysis (details in Wolfsgruber et al.[63]). The factor analysis yielded five factors for different cognitive domains: learning and memory, language, visuospatial abilities, executive function and working memory. These were further combined to a domain-general global cognitive factor in the form of their mean value. The PACC5 scores were calculated as a mean of the PACC5's five subitems (see Papp et al.[62] for a specification of the subtests). Prior to calculation of the PACC5 scores, the five subitems were z-transformed using the mean and standard deviation of the cognitively unimpaired sample consisting of CN, ADR and SCD participants. Nine of the fMRI participants lacked PACC5 test scores. One of them also had missing factor scores. In terms of the PACC5 score, for most of the subjects data from multiple time points was available (68, 125, 72, 107, 96, 17 participants with 1, 2, 3, 4, 5, 6 time points, respectively), which was used to model the cognitive trajectories longitudinally.

### CSF measures

AD biomarkers were determined using commercially available kits according to vendor specifications: V-PLEX $A\beta_{42:40}$ Peptide Panel 1 (6E10) Kit (K15200E) and V-PLEX Human Total Tau Kit (K151LAE) (Mesoscale Diagnostics LLC, Rockville, USA), and Innotest Phospho-Tau(181P) (81581; Fujirebio Germany GmbH, Hannover, Germany). Here, we focused on $A\beta_{42:40}$ and phospho-tau181 (p-tau) as CSF measures of amyloid-β and tau pathology. Of note, these were used as continuous measures.

### MRI acquisition

MRI data was acquired with Siemens scanners (3 TIM Trio systems, 4 Verio systems, one Skyra and one Prisma system) at 10 different scanning sites. The current analysis was performed using T1-weighted images (3D GRAPPA PAT 2, 1 mm³ isotropic, 256 × 256 px, 192 sagittal slices, TR 2500 ms, TE 4.33 ms, TI 1100 ms, FA 7°, ca. 5 min), T2-weighted images (optimized for medial temporal lobe volumetry, 0.5 × 0.5 × 1.5 mm³, 384 × 384 px, 64 slices orthogonal to the hippocampal long axis, TR 3500 ms, TE 353 ms, ca. 12 min) and a task fMRI protocol (2D EPI, GRAPPA PAT 2, 3.5 × 3.5 × 3.5 mm³ isotropic, 64 × 64

px, 47 slices, oblique axial/AC-PC aligned, TR 2580 ms, TE 30 ms, FA 80°, 206 volumes, ca. 9 min). For more details, see previous publications[59,64]. For task fMRI, all sites used the same 30-inch MR-compatible LCD screen (Medres Optostim) matched for distance, luminance, color and contrast constant across sites, and the same response buttons (CurrentDesign). All participants underwent vision correction with MR-compatible goggles (MediGlasses, Cambridge Research Systems) according to the same standard operating procedures. SOPs, quality assurance and assessment were provided and supervised by the DZNE imaging network.

Subjects performed a modified version of an incidental visual encoding task using pictures of indoor and outdoor scenes[54,64]. After familiarization with two so-called Master scenes (one indoor, one outdoor) outside of the scanner, participants were presented with 44 repetitions of the Master scenes (22/22) and 88 novel scenes (half outdoor, half indoor) in the MRI scanner using the software Presentation (Neurobehavioral Systems Inc.). Participants were instructed to classify each scene as either indoor or outdoor by pressing a button. Each scene presentation lasted 2500 ms, with an optimized inter-trial jitter for statistical efficiency. After a retention delay of 60 min, memory was tested outside of the scanner with a 5-point recognition-confidence rating for the 88 former novel scenes and 44 new distractor scenes, to assess successful incidental memory encoding. A response of 1 referred to "I am sure I have not seen this picture before", a 5 meant "I have definitely seen this picture before" and 3 referred to "I don't know".

### Image processing

FreeSurfer 6.0 (http://surfer.nmr.mgh.harvard.edu/) was used to obtain measures for hippocampal volumes by combining T1- and T2-weighted images using a multispectral analysis algorithm[65]. Mean cortical thickness was also acquired via FreeSurfer 6.0. Total intracranial volumes (TIV) were derived using the CAT12 toolbox (version 12.6)[66] in SPM12 r7771 (Wellcome Center for Human Neuroimaging, University College London, UK). Total gray matter volumes were calculated as cumulative sums of the gray matter probability maps from SPM segmentation (see step 3 of fMRI data processing).

fMRI data processing and analysis were performed using SPM12 and Matlab_R2016b/Matlab_R2018a. The image preprocessing followed standard procedures: (1) Slice time correction; (2) realignment and unwarping using voxel-displacement maps derived from the fieldmaps; (3) segmentation into gray matter, white matter and CSF; (4) coregistration of functional images to the structural; (5) normalization of the functional images to a population standard space via geodesic shooting nonlinear image registration; (6) normalization to MNI space via an affine transformation; (7) spatial smoothing of the functional images with a 6-mm isotropic Gaussian kernel.

In this study, we focused on reserve patterns based on the so-called subsequent memory effect, also referred to as successful (memory) encoding, which considers the BOLD-activation during encoding of a stimulus as a function of its subsequent remembering. Following recent methodological research, we decided to model the subsequent memory effect parametrically (see Soch et al.[23]) as opposed to categorically. Higher beta values of the subsequent memory contrast images indicate a stronger modulation of the local voxel-based BOLD signal according to the form of the parametric modulator (here arcsine; see below), i.e., a larger difference in BOLD during encoding of later remembered compared to neutral or later forgotten stimuli.

In the first-level general linear model (GLM), all novel scenes were collected into a single onset regressor and a parametric modulator with an arcsine-transformation was applied, resulting in the subsequent memory regressor: arcsine $\left(\frac{x-3}{2}\right) \cdot \frac{2}{\pi}$ for a given confidence rating $x$. A previous study has revealed evidence that this parametric modulator, which puts higher weights on definitely forgotten (1) or

remembered (5) items in comparison to probably forgotten (2) or remembered (4) items, is the best choice for a theoretically derived parametric modulator in the same task-design[23]. The first-level GLM further included the onsets of the Master scenes and covariates, including the six motion regressors from the realignment and a CSF-based nuisance regressor. Including nuisance regressors from regions with noise/artifact signal has been shown to increase the sensitivity of BOLD-fMRI studies[67]. In order to obtain a time series for the CSF nuisance regressor, the first eigenvariate of the BOLD time series was extracted from an anatomical CSF mask. The CSF mask was obtained by thresholding the MNI shoot template of CSF tissue probabilities with a conservative value of 0.9 and eroding it once.

Additional smoothing with a 6-mm Gaussian kernel was applied to the subsequent memory contrast images to improve the signal-to-noise ratio in the heterogeneous large clinical sample. In view of the multivariate setting of our analysis and the required dimensionality reduction of the (high-dimensional) memory contrast images, inclusion of potential noise components seemed particularly problematic. Hence, we focus on regions with significant subsequent memory contrast activation and deactivation ($p_{FWE} < 0.05$; illustrated in Fig. 2A; 13695 voxels) and therefore excluded regions that might reflect more substantial noise. The obtained task-active mask was used to restrict all subsequent fMRI-based reserve analyses.

To enhance the signal-to-noise ratio, we opted for stringent outlier exclusion criteria, predicated on behavioral and task-related fMRI metrics. Individuals were excluded if either of the following was true: (1) Errors in the indoor/outdoor judgment >8. (2) Absolute response bias >1.5 in their confidence rating. (3) Framewise displacement (FD), calculated as mean absolute difference in the six head motion parameters between subsequent EPIs[68], was above 0.5 mm in a single EPI or above 0.2 mm in more than 2% of the EPIs. (4) An individual had extreme outliers (1st quartile - 3*IQR or 3rd quartile + 3*IQR) in the beta values of more than 10% of the voxels of their (GM-masked) regressor image. 68 individuals were excluded based on these criteria, leaving an fMRI sample with 490 individuals. More comprehensive details regarding this selection process can be found in the Supplementary methods.

### One-dimensional pathological load score

We base our multivariate reserve model of CR on a dimensional approach to individual pathological load. More specifically, we here extend the ATN classification system[69] to continuous measures by focusing on joint variation across A, T and N simultaneously. This enables a simplified biological assessment of the individual pathological state, which is likely to be on the continuum from healthy to AD and also avoids difficult a-priori choices for cut-offs. The utilized ATN measures were the following: CSF Aβ$_{42:40}$ ratio (A), CSF p-tau (T) and hippocampal volumes (N). The latter were represented by the sum of their bilateral volumes, divided by the subject's TIV.

All three variables were normalized with their respective means and standard deviations to ensure similar scaling. Due to the potential nonlinearity of the disease progression trajectory along the AD continuum in 3D ATN space, a nonlinear dimensionality reduction method called t-distributed stochastic neighbor embedding (t-SNE)[70] was employed in order to reduce the dimension to one, yielding a single PL score per subject (for more details, see Supplementary information). For this purpose, the scikit-learn library 0.23.2 in Python 3.7 was utilized. The resulting PL score was normalized to range from 0 (minimal AD pathology) to 1 (maximal AD pathology in the sample).

### Multivariate reserve model of brain activity patterns

According to the recent consensus definition, a network that underlies CR should moderate the effect of brain pathologies on cognitive performance[3]. The examination of this moderation effect represents the essence of our multivariate model of CR. More specifically, we

further study CR in the context of fMRI activity patterns during memory encoding as represented by first-level GLM contrast images, one per subject, quantifying their encoding success (for details, see "Image processing" section).

First, if one assumes (scalar) brain activity in a region is given by $A$ and pathological load by PL, then a trivial (linear) moderation model that enables testing a CR effect of activity on cognitive outcomes $y$ could be described as

$$y = b_0 + b_1 \cdot A + f(A) \cdot \text{PL} + \epsilon \qquad (1)$$

with intercept $b_0$, main effect of activity $b_1$ and some function $f(A)$. For CR to improve performance, there might be (1) a (linear) additive effect of activity (e.g., $b_1 > 0$ for activations) and/or (2) a (per se nonlinear/ multiplicative) moderation effect where pathology affects cognition in terms of the slope of PL being a function of activity ($f(A) \neq const$). In principle, it would follow that regional brain activity could (by means of intervention or individual predispositions) be optimized with respect to showing improved performance and/or minimize the detrimental effect of pathology. We are aware that most biological processes are more complex, but for reasons of simplicity, we here further focus on the case where the above slope is a simple linear function of activity, i.e., $f(A) = b_2 + b_3 \cdot A$ with main effect of pathology $b_2$ and interaction/moderation effect $b_3$. Please note that in what follows, PL is just used as a quadratic term, as it has been identified as a better predictor of PACC5 compared to a linear term (see "Pathological load is associated with cognitive performance" section).

Second, it would be feasible to implement this approach in a mass-univariate (voxel-based) manner that enables testing whether a region (in isolation) contributes to CR in above described ways (1) and/or (2). However, since the subsequent memory contrast activity represents spatially correlated patterns in many brain areas reflecting distributed information processing we opt for a multivariate approach (also avoiding multiple testing and increasing sensitivity). We therefore further assume the above activity A that might contribute to CR (via $f(A)$) to be reflected by patterns of voxelwise subsequent memory contrast images (in task-active areas), i.e., $b_3 \cdot A = \sum w_i \beta_i$ with linear (group-level) weights $w_i$ describing a voxel's potential contribution to CR and its contrast value $\beta_i$. Please note that we assumed free weight parameters to be positive or negative, enabling potentially enhanced and reduced activations serving reserve processes. This approach generalizes the above ideas of univariate CR as well as brain-based multivariate cognition-prediction models by asking if there is any activity pattern (which a subject could more or less express) that facilitates CR by means of a moderation of pathology effects.

Third, due to the large number of parameters ($w_i$) we implement the multivariate reserve model by means of representing the subsequent memory contrast images by projections on P-order principal components basis functions (images) obtained from PCA. This resembles an application of principle components regression with principal components being used for quantification of patterns of (1) main effects as well as (2) the moderation effect representing CR in a narrower sense. The finally applied multivariate reserve model is a prediction model of cognitive performance, including main effects of activity patterns and their interactions with pathology:

$$y = b_0 + \sum_{p=1}^{P} b_{1,p} \cdot PC_p + \left( b_2 + \sum_{p=1}^{P} b_{3,p} \cdot PC_p \right) \cdot PL^2 + c \cdot COV + \epsilon \qquad (2)$$

with PACC5 cognitive performance scores $y$, individual pathological load score $PL$, component scores $PC_p$ for corresponding (PCA) eigen-images $p$ and COV representing the covariates age at baseline, sex, TIV and multiple binary dummy variables indicating MRI acquisition at a specific site. Since PL scores were dependent on the availability of CSF

measures, the model was restricted to a subsample of 232 participants (see Fig. S3). Age and TIV were mean-centered. Education was deliberately not chosen as a covariate in this context due to its role as a CR proxy. PACC5 scores were transformed with a Box-Cox transformation (lambda = 2.8) in order to achieve a closer approximation of the model's residuals to the normal distribution. The coefficients $b_{3,p}$ represent the moderation effect indicative of CR according to the consensus framework[3]. The optimal number of principal components $P$ required to characterize reserve patterns based on subsequent memory contrast images is a free hyperparameter in the multivariate reserve model. It was optimized using a 10-fold cross-validation approach described in the Supplementary information. In the next step, PCA was performed on the complete (mean-centered) functional data using the optimized value of $P$. The multivariate reserve model (Eq. (2)) with the previously identified optimal number of principal components was estimated on the whole data set, obtaining coefficients for each principal component. While the model was not cross-validated during model fitting, validation occurs in later stages in multiple forms (see "Validation of the CR score" section). We additionally note that our aim was not to build a predictive (AI) model with a primary focus on predictive capabilities for new data but to build an explanatory model that helps to elucidate the neural implementation of cognitive reserve (see Shmueli[71] for a comparison).

Then, the approach enables us to project the obtained moderation coefficients $b_{3,p}$ for the PCs back into the image space for the purpose of illustration and to determine the net moderation effect $w_i = \sum_{p=1}^{P} b_{3,p} V_{p,i}$ of all voxels $i$ with eigen-images $V_p$ obtained from PCA. Therefore, $w_i$ represents how (strong) the local subsequent memory contrast in voxel $i$ (i.e., the activity associated with successful memory encoding) contributes to moderation of the effect that pathology has on cognitive performance differences and thus its potential role for cognitive reserve. Finally, we introduce a useful reserve score as the amount of how an individual's successful encoding activity aligns with the reserve pattern we identified on the group level by aggregating individual contrast images using the reserve weights over all voxels in the mask: $CR_{score} = \sum w_i \beta_i$.

## Statistical analyses

**PL score.** For validation purposes, the association between the retrieved PL scores and PACC5 cognitive test scores was examined, including education and the covariates age at baseline, sex and site of data acquisition. Both models with PL as a linear and a higher-order quadratic predictor of PACC5 were tested and their performance was compared in terms of their explained variance ($R^2$ value). The quadratic predictor was tested due to visual indications for a quadratic relationship between PL and PACC5. Furthermore, such quadratic relationships have been observed in similar contexts, for example, between age and brain structure (e.g., Ziegler et al.[72]). Instead of testing a full quadratic model including a linear term, we restricted ourselves to finding a single predictor of disease severity in order to avoid a further increase in the complexity of the subsequent multivariate reserve model and thus aid its interpretability. Furthermore, the inclusion of an additional linear term did not provide substantial increases in explained variance ($\Delta R^2 = 0.014$; in comparison: $\Delta R^2 = 0.040$ between the quadratic-only and linear-only model). An additional model assessed an interaction of the PL score with education as a CR proxy according to the assumption that cognitive reserve moderates the effect that brain pathology (PL) has on cognitive outcomes (PACC5).

**Multivariate reserve model—voxel-wise inference.** Inference on the voxel-level in the context of the multivariate moderation analysis was performed using a bootstrapping procedure. The following steps were done in 5000 iterations of bootstrapping:

1. Create a bootstrap sample of equal size as the original sample used in the multivariate moderation model by randomly resampling subjects with replacement from it.
2. Estimate bootstrap coefficients $\hat{b}_{3,p}$ from Eq. (2) for the bootstrap sample.
3. Obtain individual voxel bootstrap moderation coefficients $\hat{w}_i$.

For every voxel, the coefficients $\hat{w}_i$ were then sorted in ascending order. 95% confidence intervals were obtained by specifying the lower bound as the 126th value (=2.5th percentile + 1) and the upper bound as the 4875th value (=97.5th percentile). Voxels whose 95% confidence intervals did not contain 0 were then judged as significant. Apart from inferring significance, this allowed us to estimate uncertainty in the voxels' coefficients. An alternative approach similar to the one previously presented in the context of multivariate mediation analysis of Chen et al.[73] is presented in the Supplementary information.

**Validation of the CR score**. In order to ensure the validity of the CR score (see "Multivariate reserve model of brain activity patterns" section), its moderating effect between the PL score and cognitive performance was tested in a separate moderation model using different cognitive scores on top of PACC5.

$$Performance = b_0 + b_1 \cdot BAE + b_2 \cdot PL^2 + b_3 \cdot CR_{score} \cdot PL^2 + c \cdot COV + \epsilon$$
(3)

where $CR_{score}$ represents the subject-level weighted sum of moderation coefficients ($b_{3,p}$). Moreover, $BAE$ reflects the simpler additive effect of brain activity on performance, i.e., a score calculated analogously to the CR score but aggregating the additive components $b_{1,p}$ from Eq. (2) instead. In addition to PACC5 we here used a memory factor and a global cognitive factor score as dependent variables to demonstrate that the main result obtained from learning reserve patterns based on PACC5 generalizes to other cognitive scores. This validation analysis was not possible in the subset of participants without a PL score (due to biomarker unavailablility). We instead performed a similar analysis for these participants in which the PL score (in Eq. (3)) was replaced solely by hippocampal atrophy (squared). Please note that we (imprecisely) use the term hippocampal atrophy for convenience to denominate a variable that has higher values for lower hippocampal volumes instead of actual longitudinal changes in hippocampal volumes, as the name might suggest. Hippocampal atrophy ranged from 0 to 1, like the PL score, with higher values representing smaller volumes, and was obtained by multiplying the TIV-corrected hippocampal volumes with −1 and then re-scaling them. Additionally, the correlation between the CR score and education as a typical CR proxy was assessed across the whole sample. Since the above model training and analyses were cross-sectional, as a final validation step, we utilized linear mixed-effects modeling (package lme4 in R) to test the moderation effect between pathology and the CR score longitudinally. The model included a subject-specific intercept and slope, as a model comparison had suggested a model with both random intercept and slope as superior compared to one with a random intercept alone. The model examined the three-way interaction effect between the CR score, hippocampal atrophy (squared) and time between measurements (continuous variable). It also included the corresponding two-way interactions. Hippocampal atrophy was used to maximize sample size for the longitudinal analysis. However, results for a similar model in the CSF-subsample using PL (squared) as pathology measure are presented in the Supplementary information. Age at baseline, sex, site of data acquisition, BAE and a BAE by time interaction were included as covariates. One individual was considered an extreme outlier based on their CR score (>Q3 + 3*IQR) and was thus excluded from analyses involving the CR score.

**Effect sizes**. Effect sizes of correlational analyses are presented in the form of the Pearson correlation coefficient r. They were calculated with the cor.test function of the stats package in R 4.2.2. For all regression models, standardized regression coefficients $\beta$ (change in standard deviations of a dependent variable for a one standard deviation change in the independent variable, while holding all other predictors constant) have been calculated with the functions lm.beta and stdCoef.merMod of the package lm.beta in R 4.2.2.

**Reporting summary**
Further information on research design is available in the Nature Portfolio Reporting Summary linked to this article.

## Data availability
The raw data collected in the study "DELCODE−DZNE-Longitudinal Cognitive Impairment and Dementia Study (BN012)" cannot be made openly available without violation of the data protection concept of the DZNE. The same applies to the processed individual (f)MRI images. Access to the relevant study data can be obtained by submitting an application to the Clinical Research Platform of the DZNE. The template for the application for the submission of data and biomaterial samples is available on the DZNE homepage (https://www.dzne.de/en/research/research-areas/clinical-research/databases-of-the-clinical-research/). The expected timeframe for response to access requests is 1 month. Access will be granted for 10 years. All processed data used for the analyses involving the CR score ("CR score moderates effects of pathology on cognitive performance, also longitudinally" section) are provided in Zenodo (https://doi.org/10.5281/zenodo.12820807) along with Source Data files containing all Source data and (group-level) NIfTI images in MNI space with the CR coefficients displayed in various figures (e.g., Fig. 2B)[74]. Source data are provided with this paper.

## Code availability
The code for the multivariate moderation model can be found on Github under https://github.com/znerp/NI_moderation_mv[75].

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

## Acknowledgements

We thank the members of the DELCODE group in Magdeburg for providing helpful feedback and suggestions with respect to the conducted analyses. The study was funded by the German Center for Neurodegenerative Diseases (Deutsches Zentrum für Neurodegenerative Erkrankungen (DZNE)), reference number BN012. This work was supported by the UK Medical Research Council (grant number: MR/X020274/1 to P.Z.), the National Institute on Aging (R24 AG061421 to Y.S. and G.Z.), the federal state of Saxony-Anhalt and the European Regional Development Fund (ERDF) in the Center for Behavioral Brain Sciences (CBBS, ZS/2016/04/78113 to J.M.) and the German Research Foundation (DFG, Project-ID 425899996 to E.D., A.M. and G.Z.).

## Author contributions

N.V., A.M. and G.Z. conceptualized the present data analysis. Overall design, implementation and collection of data for the DELCODE study at the different study sites was provided by E.I.I., H.S., O.P., D.Gr., L.S.S., L.P., J.P., E.J.S., S.A., A.Sch., K.F., J.W., A.R., W.G., S.T., I.K., D.Go., C.L., M.H.M., P.D., S.H., K.S., B.H.S., F.J. and E.D. L.K., E.I.I., J.B., R.Y., A.Sp., N.R., M.T.H., F.B., M.W., S.W., L.D. and E.D. provided core methodological data. J.M., E.I.I., J.B., R.Y., P.Z., N.V. and G.Z. preprocessed the MRI data. N.V. performed the statistical analyses, the present methodology (PL score, multivariate moderation model including cross-validation and PCA, CR score) and visualizations. N.V. wrote the initial draft of the manuscript. J.M., L.K., A.N., E.I.I., R.Y., M.W., P.Z., Y.S., B.H.S., E.D., A.M. and G.Z. reviewed and edited the manuscript. A.M. and G.Z. supervised the present project equally.

## Funding

## Competing interests

The authors declare the following competing interests: Y.S. consults for Eisai, Lilly, and Arcadia. Columbia University licenses the Dependence Scale, and in accordance with university policy, Y.S. is entitled to royalties through this license. B.H.S. is involved in clinical studies by Roche, Biogen, and Hummingbird Diagnostics, but does not receive personal funds from any of them. S.T. is member of the DSMB of the study ENVISION (Biogen). J.W. acted as a consultant for Immungenetics, Noselab, and Roboscreen. J.W. further served on a scientific advisory board for Abbott, Biogen, Boehringer Ingelheim, Lilly, Immungenetics, MSD Sharp-Dohme, Noselab, Roboscreen, and Roche. J.W. received honoraria for presentations from Beijing Yibai Science and Technology Ltd, Eisai, Gloryren, Janssen, Pfizer, Med Update GmbH, Roche, and Lilly. The remaining authors declare no competing interests.

## Additional information

Niklas Vockert [1] ✉, Judith Machts[1,2], Luca Kleineidam [3,4], Aditya Nemali[1,2], Enise I. Incesoy [1,2,5], Jose Bernal[1,2], Hartmut Schütze [1,2], Renat Yakupov [1,2], Oliver Peters[6,7], Daria Gref [7], Luisa Sophie Schneider [8], Lukas Preis[7], Josef Priller [6,9,10,11], Eike Jakob Spruth[6,9], Slawek Altenstein[6,9], Anja Schneider[3,4], Klaus Fliessbach[3,4], Jens Wiltfang [12,13,14], Ayda Rostamzadeh[3], Wenzel Glanz[1], Stefan Teipel[15,16], Ingo Kilimann[15,16], Doreen Goerss[15,16], Christoph Laske[17,18], Matthias H. Munk[17,19], Annika Spottke[3,20], Nina Roy[3], Michael T. Heneka [3,21], Frederic Brosseron [3], Michael Wagner [3,4], Steffen Wolfsgruber[3,4], Laura Dobisch[1], Peter Dechent[22], Stefan Hetzer[23], Klaus Scheffler[24], Peter Zeidman [25], Yaakov Stern[26], Björn H. Schott [12,13,27], Frank Jessen[3,28,29], Emrah Düzel[1,2], Anne Maass [1,30] ✉, Gabriel Ziegler [1,2,30] ✉, the DELCODE study group

[1]German Center for Neurodegenerative Diseases (DZNE), Magdeburg, Germany. [2]Institute of Cognitive Neurology and Dementia Research (IKND), Otto-von-Guericke University, Magdeburg, Germany. [3]German Center for Neurodegenerative Diseases (DZNE), Bonn, Germany. [4]University of Bonn Medical Center, Department of Neurodegenerative Diseases and Geriatric Psychiatry, Bonn, Germany. [5]Department for Psychiatry and Psychotherapy, University Clinic Magdeburg, Magdeburg, Germany. [6]German Center for Neurodegenerative Diseases (DZNE), Berlin, Germany. [7]Charité - Universitaetsmedizin Berlin, corporate member of Freie Universitaet Berlin and Humboldt-Universitaet zu Berlin, Institute of Psychiatry and Psychotherapy, Berlin, Germany. [8]Charité - Universitaetsmedizin Berlin, corporate member of Freie Universitaet Berlin and Humboldt-Universitaet zu Berlin, ECRC Experimental and Clinical Research Center, Berlin, Germany. [9]Department of Psychiatry and Psychotherapy, Charité Berlin, Germany. [10]School of Medicine, Technical University of Munich, Department of Psychiatry and Psychotherapy, Munich, Germany. [11]University of Edinburgh and UK DRI, Edinburgh, UK. [12]German Center for Neurodegenerative Diseases (DZNE), Goettingen, Germany. [13]Department of Psychiatry and Psychotherapy, University Medical Center Goettingen, University of Goettingen, Goettingen, Germany. [14]Neurosciences and Signaling Group, Institute of Biomedicine (iBiMED), Department of Medical Sciences, University of Aveiro, Aveiro, Portugal. [15]German Center for Neurodegenerative Diseases (DZNE), Rostock, Germany. [16]Department of Psychosomatic Medicine, Rostock University Medical Center, Rostock, Germany. [17]German Center for Neurodegenerative Diseases (DZNE), Tuebingen, Germany. [18]Section for Dementia Research, Hertie Institute for Clinical Brain Research and Department of Psychiatry and Psychotherapy, University of Tuebingen, Tuebingen, Germany. [19]Department of Psychiatry and Psychotherapy, University of Tuebingen, Tuebingen, Germany. [20]Department of Neurology, University of Bonn, Bonn, Germany. [21]Luxembourg Centre for Systems Biomedicine (LCSB), University of Luxembourg, Luxembourg, Luxembourg. [22]MR-Research in Neurosciences, Department of Cognitive Neurology, Georg-August-University Goettingen, Goettingen, Germany. [23]Berlin Center for Advanced Neuroimaging, Charité - Universitaetsmedizin Berlin, Berlin, Germany. [24]Department for Biomedical Magnetic Resonance, University of Tuebingen, Tuebingen, Germany. [25]Wellcome Centre for Human Neuroimaging, UCL Institute of Neurology, London, UK. [26]Cognitive Neuroscience Division, Department of Neurology, Columbia University Vagelos College of Physicians and Surgeons, New York, NY, USA. [27]Leibniz Institute for Neurobiology, Magdeburg, Germany. [28]Department of Psychiatry, University of Cologne, Koeln, Germany. [29]Excellence Cluster on Cellular Stress Responses in Aging-Associated Diseases (CECAD), University of Cologne, Koeln, Germany. [30]These authors contributed equally: Anne Maass, Gabriel Ziegler. ✉e-mail: Niklas.Vockert@dzne.de; Anne.Maass@dzne.de; Gabriel.Ziegler@dzne.de

## the DELCODE study group

Enise I. Incesoy [1,2,5], Oliver Peters[6,7], Lukas Preis[7], Slawek Altenstein[6,9], Josef Priller [6,9,10,11], Eike Jakob Spruth[6,9], Frederic Brosseron [3], Klaus Fliessbach[3,4], Annika Spottke[3,20], Michael Wagner [3,4], Steffen Wolfsgruber[3,4], Björn H. Schott [12,13,27], Jens Wiltfang [12,13,14], Frank Jessen[3,28,29], Ayda Rostamzadeh[3], Laura Dobisch[1], Emrah Düzel[1,2], Wenzel Glanz[1], Renat Yakupov [1,2], Gabriel Ziegler [1,2,30] ✉, Ingo Kilimann[15,16], Stefan Teipel[15,16], Christoph Laske[17,18] & Matthias H. Munk[17,19]

