## [Peer Review File · Nature Communications]

Cognitive Reserve Against Alzheimer's Pathology Is Linked to Brain Activity During Memory FormationReviewer #1 (Remarks to the Author):

The manuscript by Vockert and colleagues describes an analysis to determine brain functional activation patterns that underly cognitive reserve. The analysis is conducted in a large sample of healthy elderly individuals and individuals across the Alzheimer's disease (AD) spectrum, each of whom underwent a scene memory encoding task during functional MRI (fMRI) scanning, and about half of whom also have CSF biomarkers indicating presence (or absence) and degree of Alzheimer's disease neuropathology. Neuropathology is summarized in a single combinatorial measure normalizing nonlinear advancement of three canonical AD biomarkers into a single linear score. Cognitive reserve is measured as the interaction between neuropathology and functional activation in predicting task performance. The authors thus identify functional activation patterns that signify preserved cognitive performance despite elevated AD neuropathology. The main findings suggest that individuals harnessing AD pathology that are nonetheless able to recruit normative task-specific activation patterns are able to achieve better task performance. The authors use this normative task-specific activation pattern as an individual-level proxy for CR, which was found to be weakly but significantly correlated with education (a common CR proxy) at the individual level. The authors show that individuals demonstrating this activation pattern also have preserved cognition on other cognitive tests, and slower longitudinal cognitive decline. The authors thus interpret this activation profile as a CR-related phenomenon, which they discuss in detail.

This is a thorough and well-written study that is poised to be a needle-moving and timely contribution to the rich cognitive reserve literature. There are many strengths of the study, including careful adherence to consensus recommendations on CR research, an impressive sized and well described cohort of elderly individuals with fMRI data, and a focus on well-controlled experimental activation of in vivo memory circuits. The methods are meticulously documented and most decisions are justified. While the potential of this study is quite high, there remain lingering methodological flaws and issues with interpretations that hamper enthusiasm for the manuscript in its current form. These limitations are detailed below:

MAJOR CONCERNS

1) P14, line 292 "This indicates that some individuals are able to maintain functional integrity in parts of the core cognitive circuitry despite the presence of AD pathology." I think this very well sums up the main findings of this paper. However, this is also perhaps not an earth-shattering finding. Individuals maintaining successful task-specific activation are more successful at the task, pathology or not. I'm not sure this gets us any closer to understanding CR mechanisms. A more pressing question is *why* certain participants are able to maintain successful task-specific activation. Is there an increase in global signal or blood flow? Is there a region or network mediating this moderation? Are there characteristics at rest that predict successful activation? The authors theorize the ability to maintain successful DMN deactivation, which is reasonable considering burgeoning literature pointing to circuit hyperactivation in AD. Can the authors show this with the data they have available?

2) Perhaps related to the above, the authors use the whole-sample encoding mask in their CR score. But since they are reducing the data dimensions anyway, why not extend the search-space to the whole-brain? The findings that the CR regions resemble the memory encoding regions is a bit circular given that the search space was limited to these regions anyway.

Couldn't regions outside of this mask contribute specifically in the case of pathology, but not in normative task activation? And wouldn't that be interesting (and perhaps more aligned with traditional thoughts on CR representing compensatory functional re-organization)?

3) Moderating variables can at times also be mediating variables, and in this case it may be important to know because it would change interpretations a bit. Some evidence that this might be occurring is that there seems to be a main effect of CR on the intercept in Figure 6B (would this be expected?). It would perhaps help to see, in Figure 5, the dots colored by activation so as to see whether individuals with high PL frequently have strong CR-related activation. Also, a formal mediation should be tested.

4) The authors repeatedly signal the validation of the CR measure in their study (repeated in the abstract, discussion, etc), but the generalizability of the CR measure is actually not all that clear to me at present. Consider:

a) The only cross-validation used in model fitting is in determining the optimal number of PCs to reduce the data for the CR estimates. However, this "hyperparameter" doesn't seem to have a big impact on the model. The confidence intervals are huge and the values are barely changing (Fig S5). Would the results change substantially if using e.g. just two PCs? Doing so may actually be to the authors benefit regarding the next point...

b) In fact, the model weights do not appear to be cross-validated (at least during model fitting). These weights are ultimately what would allow this model to be generalized to new data. Whether this actually occurs later in the manuscript is not entirely clear. If not, there's no real way of testing whether the authors are overfitting to their data here.

c) While less important, it should be noted that the authors choose the best model by looking at the mean across folds. This is not correct. Instead, one should look at performance (loss) measured on all out-of-sample data in aggregate. This allows the performance to be estimated across the same amount of data that is observed, while still being all left-out data. It should also be stated what the loss function was? If R-squared, was that literally the square of the r-value (not optimal since it is agnostic to direction) or the actual coefficient of determination?

d) P12, L247: "in an analysis encompassing the remaining sample lacking a PL score (due to missing CSF data), a weaker yet significant moderation effect of the CR score on the association between hippocampal volumes and cognitive performance was observed in the form of the latent memory factor ($p = 0.018$, $\beta = 0.131$) and PACC5 ($p = 0.011$, $\beta = 0.145$)." This is perhaps the most convincing and interesting finding in the paper since it may actually indicate model generalization. This is not clear, though, since it is unclear whether this analysis used the parameters directly from the trained model, or whether the model was refit on the new participants using HV instead of PL score? If the former, point 4B above would be mostly satisfied but, if the latter, this analysis is hampered by the same flaw. Either way, this analysis should be visualized. Note that this finding is more interesting (to me) than "validation" with other cognitive tests and cognitive decline in the same subjects, since these factors all tend to correlate pretty strongly (and with task performance) in AD-spectrum samples, so validation would be expected.

5) I like the approach to summarize AD pathology with the PL score, though it does come with

some limitations. The supplemental data shows the effect of AD pathology on the PL score is quite non-linear. For example, a PL score between 0.4 and 0.6 seems to involve lowish HV and highish p-tau, but normalish ab42-40 levels. However, a score between 0.6 and 0.8 indicates abnormal CSF biomarkers but fairly preserved HV. I would not really consider someone in the first category to be “less progressed” than the second category (especially as it pertains to cognition), though they might be more AD-like. In addition, the reason all the results require quadratic function is likely because the relationship between PL score and HV is also quadratic, and HV is probably driving most of the effects in the paper. This seems like a limitation of using tSNE for this purpose, which really isn’t well designed for what the authors are trying to accomplish. Many such approaches have been described and compared (<https://www.nature.com/articles/s41587-019-0071-9>) that might better suit the authors’ purposes (I wonder if even a simple first eigenvalue approach would do better?).

OTHER METHODOLOGICAL QUESTIONS

6) The reported collinearity between CR activation and basic encoding activation calls to question which is driving the effect in the “validation” analyses. If the authors substitute the encoding parameter (main effect) instead of the CR effect into these models, or puts them both in the model, what happens?

7) Only some individuals have CSF biomarkers. It would be helpful to know if there was any bias in which participants received both MRI and CSF vs. which received only MRI? What explains this discrepancy and could it be relevant to the study findings?

8) The authors should state explicitly somewhere prominent (e.g. Results) that their QC procedure disproportionately removed cognitively impaired individuals — only 14% of subjects were removed, but this included over half of AD patients and almost a third of MCI patients.

9) It is confusing that the the authors use PCs to summarize the CR data, but then later break the results into clusters. Are these clusters distinct from the PCs?

10) In Figure 5 — why aren’t there confidence intervals? In Figure 6B — why do the number of timepoints only go up to 5, and why does it seem like everyone has 5 timepoints (the methods stated up to 6 timepoints but not everyone had all timepoints)? Why does it seem like only two points are visualized?

A FEW LINGERING MINOR POINTS

* P19 line 482 “Participants were classified as MCI when displaying an age-, sex-, and education-adjusted performance below -1.5 SD on the delayed recall trial of the CERAD word-list episodic memory tests” — would it be fair to call these subjects amnesic MCI?

* For CSF analysis, while full details are not necessary, please at least record which approach/lab/company processed the data

* Please visualize the CR vs education score. Beta and p-values alone don’t tell us much, it

helps to see the data and there is plenty of room in the supplementary

* The sentence on P15 that starts at Line 312 seems completely out of nowhere. It seems to cite a poster or something, and is probably not the best example of the point the authors are making considering other published work out there.

* Fig 2A images would be easier to ascertain if projected to a cortical surface, as in 3B. But please in these (and those in 3B) include a medial view. There are now ample tools for quickly and easily plotting volumetric data onto surfaces

Reviewer #2 (Remarks to the Author):

This is an excellent, very well-written and scientifically rigorous paper investigating the neural implementation of Cognitive Reserve (CR) across the Alzheimer's clinical spectrum utilizing task-related fMRI and A/T/N biomarkers. The work has several strengths - I'd emphasize its robust conceptual framework, the longitudinal validation of the approach, and the meticulous justification of each methodological choice.

I only have a few questions and comments:

1- The neural implementation of CR may vary across disease stages. While I understand that this is, to some extent, accounted for as PL is continuously measured across the disease, it would be interesting to discern whether the implementation of CR remains consistent between cognitively normal, SCD, MCI or AD patients or - if sample size does not allow - between non-demented and demented individuals. Could the authors calculate and project the moderations coefficients (i.e., CR pattern) within those groups?

2- Similarly, given the increasing interest and acknowledgment of sex/gender differences in reserve and resilience, it'd be interesting to know whether the CR pattern is different in males and females. This could be presented a supplemental material.

3- If my understanding is accurate (line 598) all the analyses were restricted to the task-related fMRI activation mask. If so, the potential engagement of additional areas remains unexplored. Consequently, the claims regarding cognitive reserve being implemented as a sustained recruitment of core cognitive circuits (as opposed to the activation of alternative regions) may need qualification. Could you please clarify this aspect?

4- I was surprised of the choice of creating a PL score that incorporated both Alzheimer's disease pathologies and hippocampal volume. I understand the rationale behind representing A/T/N. However, hippocampus is involved in memory encoding and higher hippocampal volume may potentially serve as a measure of both disease progression (neurodegeneration of atrophy) and of brain reserve as it may be influenced the exposure to protective factors (eg. exercise). I wonder whether this methodological decision has impacted the results and explains some of the discordant results within the hippocampus. Could the authors clarify whether the results are similar when only amyloid and tau are considered?

5- The authors discuss various potential neural mechanisms underlying the identified CR

activation pattern, including vascular supply, clearance, inflammation, and compensation by non-affected neural populations within the cognitive reserve circuit. Examining the structural integrity, such as brain volumes in the CR-related regions, could provide insights into this latter aspect.

Minor:

1. The authors indicate the existence of 51 first-degree relatives of Alzheimer's Disease (AD) patients. Could you provide clarification on whether all participants were assessed for a history of sporadic AD, and if so, how was this assessed?

2. Line 176 contains a typo it should read “as well as” instead of “as well”

3. Lines 204-206: The authors may want to check the work by Elman and colleagues doi: 10.1038/nn.3806

4. Could you please confirm – and provide clarification – that the dimensional reduction for PL calculation, considers the distinct directions of the variables where CSF A β and hippocampus volume (less is worse), and CSF tau (more is worse)?

5. Line 366: the reference to the results by Franzemeier et al. identifying the left prefrontal cortex (when discussing findings on the fusiform, temporal and occipital regions) seem to lack specific discussion. Were the authors surprised by the absence of findings in the left prefrontal cortex?

Reviewer #3 (Remarks to the Author):

The present study examined whether patterns of brain activity during memory formation may constitute a neural basis of cognitive reserve (CR) as defined by recent research criteria. They found that a multivariate pattern of activity related to the subsequent memory (SM) effect moderated the cross-sectional association between a multivariate pathology score and cognitive performance. They then generated subject-specific values reflecting the degree to which an individual's encoding activity resembled this pattern and found that higher similarity was associated with less longitudinal decline. Higher similarity scores were also correlated with education, a commonly used measure of cognitive reserve.

Overall, this is a well-written manuscript with the potential to contribute novel insight into neural mechanisms that underlie resilience to AD-related processes. The authors present strong conceptual grounding to motivate their study, using a definition of CR that focuses on how a given measure moderates the relationship between pathology and cognition. The multivariate methods proposed here may represent a useful approach to studying CR in other contexts. There are however some key conceptual and methodological questions that remain, and the manuscript would benefit from their clarification. For example, results from longitudinal analysis can provide strong support for CR, but the model as currently specified may be just as likely to detect differences in disease progression/staging as differences in CR. These issues are detailed below, as well as some minor suggestions.

Major Comments:

- In looking at the variance explained by the principal components, it seems that the first 2 PCs may be sufficient to capture the bulk of the variance. Does an analysis using this more parsimonious basis set perform similarly?
- Related to the above, does a similarity score for each individual relative to the group SM contrast (or an SM contrast within cognitively normal only) produce similar results? If so, it could simplify interpretations (e.g., higher/lower activity may be easier for readers to understand than voxel or subject loadings summed across multiple PCs). There were discordant regions between the CR-related map and SM map, indicating the maps are not completely redundant, but it is unclear how much of an impact these discrepancies would have on the results.
- Higher similarity scores are interpreted as individuals with high CR, but an alternative is that lower similarity scores reflect individuals with more progressed disease. Although the PL score controls for hippocampal atrophy, it does not rule out individual differences in overall atrophy. As an example of this alternative explanation, the panels of Figure 6B could be reinterpreted as showing quartiles of disease staging rather than CR. Individuals in quartile 4 may be considered to have lower resilience, or they may just be further along in the disease process, with more widespread atrophy causing disruptions to encoding activity and lower cognitive performance. The possibility that measures of CR are recapitulating measures of disease progression is in no way specific to this study – it is difficult to rule out in many studies of CR. However, it should be addressed, whether through further analysis or in the discussion.
- It is interesting that the voxels contributing to the CR effect seem to be spared from atrophy. They may be contributing to CR by compensating for regions that are more impacted by pathology/atrophy. Perhaps this also presents an opportunity to address the previous review comment? Models that control for the extent of cortical atrophy would indicate that differences in brain activity are not simply reflecting a greater extent of neurodegeneration. This could be done with voxelwise maps as covariates, or even with a summary metric indicating the extent of cortical neurodegeneration.
- The cross-sectional analysis focuses on the moderation effect of the CR pattern, consistent with the definition of CR given in the Introduction. However, the longitudinal analysis does not include a similar moderation effect on pathology. There were other discrepancies between cross-sectional and longitudinal models that should also be resolved or explained, such as the inclusion of BAE and diagnostic group as covariates.

Minor revision/editing/language suggestions:

- Lines 4-5 “tested this hypothesis in the Alzheimer’s disease continuum” – the sample description appears to include a majority of individuals that are not necessarily on the AD continuum unless there is evidence of abnormal biomarkers (152 CN, 51 participants with relatives with AD, and 202 participants with subjective cognitive decline).
- Line 17 “as the primary mechanism of CR” – one possible mechanism of CR was tested in this study, and results are unable to determine if it is the primary mechanism of CR.
- Lines 140-142: “which suggested the pivotal role of education in shaping how AD pathology influences cognitive abilities” implies causal associations based on a correlational analysis.
- Lines 157-159, 170: “patterns of brain regions contributing both positively or negatively to the moderation of AD pathology” – may be more accurate to state ‘to the moderation of the relationship between AD pathology and cognitive performance’
- Please clarify the following lines 212-215: “Taken together, the correlation between both patterns stood at 0.384, underlining that predominantly more of the typical i.e.

activation/deactivation can support cognitive functioning while region-specific multifaceted relationships between these neural signatures and cognitive reserve might exist.”

- Lines 223-224: Instead of “high levels of brain activity”, what about “higher SM contrast values? This would capture both higher and lower activity related to subsequent memory.
- It would be useful to provide some discussion around the alternate and statistically equivalent interpretation of the main moderation findings, i.e., rather than higher CR scores moderating the relationship between AD-related pathology and cognitive performance, higher pathology scores moderate the relationship between CR scores and cognitive performance.
- Please clarify what the p-values in lines 242-245 correspond to.

Response Letter: NCOMMS-23-49941A

We thank the editor and reviewers for their in-depth constructive comments and for the helpful suggestions. We believe that the revisions made in response to the reviewers' valuable feedback have significantly strengthened the manuscript in all sections. These changes have not only addressed the concerns raised but have also enriched the clarity and depth of our findings. Here is an overview of the comprehensive changes that have been made and constructive analyses that have been added.

First, we have implemented a new inference method via bootstrapping, replacing the method by Chén et al. (2018) in order to provide local estimations of confidence and circumvent problems with violated assumptions in specific requested analyses. This had minor implications for most figures and tables, which have been adjusted accordingly. The longitudinal validation model of the cognitive reserve (CR) score has been extended to include a three-way interaction between the CR score, pathology and time in order to eliminate discrepancies between the consensus framework by Stern et al. and our implementation. Other discrepancies between models were rectified by converting hippocampal volumes to hippocampal atrophy (ranging from 0 to 1, as the PL score), also improving statistics in the validation analyses. As requested, we explored multiple alternative dimensionality reduction methods of the AD biomarker data as well as their implications for the results. For three of them the relationship of the PL scores to AD biomarkers and the corresponding CR-related activity patterns from a multivariate model with these PL scores are presented in the supplementary data. To explore the impact of dimensionality reduction, the multivariate model has been recalculated for 2-9 principal components (PCs) and the relationship of their results as well as overlapping patterns are presented in an additional supplementary figure. For 2 PCs, the CR-related activity pattern is explicitly shown in a supplementary figure and a derived CR score was tested in cross-sectional and longitudinal validation analyses provided in the supplementary. Moreover, CR-related activity patterns have been considered across different subpopulations and disease stages in the supplementary. To verify the independence of the results on brain reserve via retained brain structure, additional analyses were included that account for morphometric covariates in both the multivariate moderation model and validation models of the CR score. We further compared the CR score to a simpler similarity-based score and evaluated its performance in validation analyses in the supplementary. In an attempt to better understand why certain participants are able to maintain activation patterns, we investigated multiple potential predictors and contributors of the CR score (including resting state-fMRI connectivity measures). Finally, we performed a mediation analysis to test the hypothesis that CR mediates the effects of the PL score on cognitive performance, despite the focus of our manuscript on the moderation framework.

We are confident that the revised manuscript makes a substantial contribution to the understanding of cognitive reserve.

REVIEWER COMMENTS

Reviewer #1 (Remarks to the Author):

The manuscript by Vockert and colleagues describes an analysis to determine brain functional activation patterns that underly cognitive reserve. The analysis is conducted in a large sample of healthy elderly individuals and individuals across the Alzheimer's disease (AD) spectrum, each of whom underwent a scene memory encoding task during functional MRI (fMRI) scanning, and about half of whom also have CSF biomarkers indicating presence (or absence) and degree of Alzheimer's disease neuropathology. Neuropathology is summarized in a single combinatorial measure normalizing nonlinear advancement of three canonical AD biomarkers into a single linear score. Cognitive reserve is measured as the interaction between neuropathology and functional activation in predicting task performance. The authors thus identify functional activation patterns that signify preserved cognitive performance despite elevated AD neuropathology. The main findings suggest that individuals harnessing AD pathology that are nonetheless able to recruit normative task-specific activation patterns are able to achieve better task performance. The authors use this normative task-specific activation pattern as an individual-level proxy for CR, which was found to be weakly but significantly correlated with education (a common CR proxy) at the individual level. The authors show that individuals demonstrating this activation pattern also have preserved cognition on other cognitive tests, and slower longitudinal cognitive decline. The authors thus interpret this activation profile as a CR-related phenomenon, which they discuss in detail.

Q1.0 This is a thorough and well-written study that is poised to be a needle-moving and timely contribution to the rich cognitive reserve literature. There are many strengths of the study, including careful adherence to consensus recommendations on CR research, an impressive sized and well described cohort of elderly individuals with fMRI data, and a focus on well-controlled experimental activation of in vivo memory circuits. The methods are meticulously documented and most decisions are justified. While the potential of this study is quite high, there remain lingering methodological flaws and issues with interpretations that hamper enthusiasm for the manuscript in its current form. These limitations are detailed below:

R1.0 We thank the reviewer for the positive assessment of our study.

MAJOR CONCERNS

Q1.1 P14, line 292 "This indicates that some individuals are able to maintain functional integrity in parts of the core cognitive circuitry despite the presence of AD pathology." I think this very well sums up the main findings of this paper. However, this is also perhaps not an earth-shattering finding. Individuals maintaining successful task-specific activation are more successful at the task, pathology or not. I'm not sure this gets us any closer to understanding CR mechanisms. A more pressing question is *why* certain participants are able to maintain successful task-specific activation. Is there an increase in global signal or blood flow? Is there a region or network mediating this moderation? Are there characteristics at rest that predict successful activation? The authors theorize the ability to

maintain successful DMN deactivation, which is reasonable considering burgeoning literature pointing to circuit hyperactivation in AD. Can the authors show this with the data they have available?

R1.1 We thank the reviewer for these helpful ideas and for encouraging us to establish a deeper understanding of these findings. In alignment with the reviewer's third comment (Q1.3), we conducted additional analyses and explored the relationships between the CR score and relevant other variables, including biomarkers of pathology. Indeed, higher CR scores were positively associated with hippocampal volumes, CSF A β ratio and, negatively, with p-tau181 levels and the PL (pathological load) score. Likewise, the CR score was found to be weakly related to log-transformed white matter hyperintensity volumes. In summary, higher CR scores were related to less AD and WM pathology. Additionally, the CR score showed weak positive correlations with structural measures such as the mean gray matter volume in CR-related regions and mean cortical thickness, indicating a significant but minor contribution of morphological differences. Nevertheless, inclusion of these structural measures (as covariates) had essentially no effect on the main results, neither in the multivariate moderation model nor in the validation models of the CR score.

As suggested by the reviewer we further performed novel analyses exploring the potential contribution of resting state-fMRI connectivity to individual differences in the CR score. Interestingly, functional connectivity in seven resting state networks (default mode, dorsal attention, fronto-parietal, limbic, somatomotor, salience, visual) were not significantly related to the CR score. Similarly, the mean global task-fMRI signal was also not related to CR score variability. As we discuss more specifically in our response to the third question (Q1.3), CR differences and pathology might be connected via a vicious cycle, in which aberrant activity patterns lead to an accumulation of more pathology (esp. A β), which in turn might disrupt activity patterns even more. Likewise, white matter hyperintensity volumes were weakly related to the CR score, suggesting that sustained vascular brain health might contribute positively to CR. Even after extensively studying various subject-level factors and incorporating additional imaging modalities (rs-fMRI and FLAIR), we did not find other characteristics significantly related to individual CR score variability than the ones mentioned above. Unfortunately, we have no available sequences for investigating a relationship with blood flow in this cohort, as suggested by the reviewer.

Finally, we would like to add two more points in response to the raised concern. First, in addition to the specific findings about a CR network, we believe that the proposed framework of pattern-level reserve analysis is a generic contribution to the field which could reveal valuable applications in future studies on reserve. This involves potentially better powered samples with different functional tasks/contrasts, multivariate resting state connectivity or even structural aspects as reserve features. This might also involve studies with a richer characterization of background factors for subsequent correlational analyses to explore its nature. Second, the empirical finding of this study, that the reserve pattern coarsely aligns also with task-specific activation patterns, might not hold for all tasks. One might speculate that, applied to different tasks, the approach could therefore give valuable insights about the existence of yet unobserved compensatory recruitment of non-task areas or functional reorganisation.

We added the following paragraph to the supplementary:

"In an attempt to understand why certain participants are able to maintain activation patterns, we investigated potential predictors and contributors of the CR score. In this sample, neither resting-state functional connectivity within seven standard networks nor mean global task-fMRI signal were

significantly related to CR score variability (see Fig. S2). However, higher CR scores were associated with less pathological measures of AD burden such as the (squared) PL score ($p = 2.84 \cdot 10^{-5}$, $r = -0.272$ [-0.387,-0.148], $df = 229$), its components $A\beta_{42:40}$ ($p = 2.93 \cdot 10^{-4}$, $r = 0.189$ [0.110,0.354], $df = 229$), p -tau ($p = 0.017$, $r = -0.166$ [-0.289,-0.037], $df = 229$) and hippocampal volume ($p = 3.65 \cdot 10^{-6}$, $r = 0.208$ [0.121,0.291], $df = 487$), as well as lower global white matter hyper-intensity volumes ($p = 0.035$, $r = -0.098$ [-0.188,-0.007], $df = 460$). The CR score was further weakly positively correlated with total GM volumes in the regions with significant contributions to CR ($p = 0.013$, $r = 0.112$ [0.024,0.199], $df = 487$) and mean cortical thickness ($p = 0.023$, $r = 0.103$ [0.015,0.190], $df = 487$). Yet, it should be noted that accounting for these structural differences as covariates had essentially no effect on the observed results, neither in the multivariate moderation model nor the validation analyses of the CR score (see Fig. S19) and Tab. S1. Taken together, although AD pathology indices, tissue volumes and white matter lesions were slightly associated with individual CR score differences, the pattern of CR-related brain areas and its predictive value for memory performance was not mediated by atrophy alone or network connectivity at rest.”

Q1.2 Perhaps related to the above, the authors use the whole-sample encoding mask in their CR score. But since they are reducing the data dimensions anyway, why not extend the search-space to the whole-brain? The findings that the CR regions resemble the memory encoding regions is a bit circular given that the search space was limited to these regions anyway. Couldn't regions outside of this mask contribute specifically in the case of pathology, but not in normative task activation? And wouldn't that be interesting (and perhaps more aligned with traditional thoughts on CR representing compensatory functional re-organization)?

R1.2 We very much appreciate the reviewer's insightful comment and suggestion. We completely agree that there are reasonable arguments to consider an even larger model input space that is not restricted to the successful encoding network. We initially constrained our multivariate reserve analysis to this particular search space, as regions that are not significantly involved in successful encoding might largely represent irreducible noise or artifacts. Our reasoning was that including larger portions of the brain / non-active areas might result in (A) even more model parameters that increase the likelihood of overfitting given the decent but not massive fMRI sample; and (B) result in small model coefficients with slightly greater uncertainty. Nevertheless, we followed the reviewer's helpful suggestion and tested the effects of extending the input space to a whole brain gray matter mask. These novel analyses were included in the Supplementary. Model fitting involved also performing novel cross-validation, in this particular case identifying four as the optimal number of principal components. Interestingly, the coefficients of voxel-contributions to the CR network obtained this way substantially correlate with the group-level successful memory encoding beta coefficients (0.942; coefficients displayed in supplementary figure S9, shown below). It appears as if such a model would only learn to identify the successful memory network, with systematically activating regions contributing to reserve, and not revealing much evidence for compensatory brain activity in other areas. At the same time, we would advise against overinterpretation about the absence of potential compensatory mechanisms. Task fMRI contrasts do show comparably high signal variability and noise contributions from different sources (see e.g. Liu et al. 2016), which is even more expected in context of multi-centric studies and clinical populations as studied here. Therefore, the observation is not unexpected that a multivariate model aiming to predict cognitive performance (using local input patterns and specific interaction terms), might also identify and separate brain areas with systematic

vs. less systematic or noisy signals. We believe that a reduction of the analysis space (for model inputs) to regions of the successful memory network as a means to reduce noise is a very reasonable choice in the light of these results. This choice would, in principle, also enable a better analysis of variations among those active areas with respect to reserve.

We have incorporated these findings in the discussion of the revised manuscript:

"Our search (or model input-) space was limited to the widespread successful encoding network, restricting our ability to identify regions that show compensatory activity to these areas. An analysis extending this search space to all gray matter indicated CR regions essentially as a resemblance of the successful encoding network (Fig. S10). However, we also advise against overinterpretation of these findings since we cannot exclude the possibility of compensatory activity in other tasks or fMRI contrasts."

Given these findings, we believe that there are reasons to choose a more cautionary wording with respect to compensatory mechanisms. Thus, we rephrased a sentence in the abstract from *"Our findings suggest maintenance of core cognitive circuits including the DMN and ACC as the primary mechanism of CR"* to *"Our findings primarily provide evidence for the maintenance of core cognitive circuits including the DMN as a mechanism of CR."*

Additionally, we changed our wording in the corresponding section of the discussion from *"In conclusion, rather than relying on the recruitment of additional brain regions as a compensatory mechanism, our findings point towards CR factors operative within core circuits themselves"* to *"In conclusion, our findings primarily support the notion that some CR factors might operate within core circuits themselves above compensatory activity discordant with successful encoding activity."* We also included a sentence to the last paragraph of the discussion that reads *"However, adequate judgment about compensation in the context of cognitive reserve should be based on further studies specifically designed for its investigation, involving multi-task and -contrast information as well as manipulation of task demand."*

Fig. S10: **CR coefficients in whole gray matter.** (A) Results of the same multivariate model when extending the search space to all gray matter instead of only regions contributing to successful memory encoding. (B) Mean beta values of the parametric successful memory contrast. The correlation between A and B is 0.942. All values have been normalized by the highest absolute value of the respective image.

Q1.3 Moderating variables can at times also be mediating variables, and in this case it may be important to know because it would change interpretations a bit. Some evidence that this might be occurring is that there seems to be a main effect of CR on the intercept in Figure 6B (would this be expected?). It would perhaps help to see, in Figure 5, the dots colored by activation so as to see whether individuals with high PL frequently have strong CR-related activation. Also, a formal mediation should be tested.

R1.3 We appreciate the interesting observation and helpful suggestion by the reviewer. As requested, we implemented the corresponding models accordingly and tested a formal mediation with the data at hand, which indicated that the CR score indeed mediates the effect of PL (squared) on PACC5 (ACME: -0.1566 [-0.3229, -0.03], $p = 0.013$). The finding that lower pathological load and our brain activity measure of CR are negatively correlated would be in accordance with the interpretation that "healthy" brain activity patterns also protect from AD pathology whereas "unhealthy" (aberrant) brain patterns promote spread of AD pathology. Alternatively, pathology could promote "unhealthy" brain activity patterns. Both options are supported by previous findings suggesting a vicious cycle in which A β /tau induces hyperactivity, which in turn leads to A β /tau accumulation.

We integrated this finding in the supplementary of the manuscript and discussed it accordingly in the main text. In the supplementary added the following result:

"Due to the relationship between PL and the CR score, we also tested a formal mediation, which indicated that the CR score mediates the effect of PL (squared) on PACC5 (average causal mediation effect: -0.1566 [-0.3229, -0.03], $p = 0.013$)."

We further address this issue in the revised discussion section:

"Besides the moderation effect, the CR score, which represents CR-related activity during successful memory encoding, was also weakly correlated with different measures of pathology and mediated the effect of PL on PACC5 scores (see supplementary). These findings point to a more intricate relationship between pathology, brain activity and cognitive performance, where low CR/hyperactivity and pathology could promote each in a vicious cycle^{31,42,43,44,45,46}."

With regards to Figure 5, we do appreciate the idea. However, we also worry that coloring the dots by CR might confuse some readers, as the colors would not be directly related to the presented regression lines. In this Figure the lines do not represent population-level predictions of the model, but rather illustrate predictions for a theoretical individual with a certain (*a-priori* specified) level of (de)activation. We revised the caption to improve clarity, which now reads as follows:

"The relationship between the PL score and the PACC5 score (Box-Cox transformed and residualized for covariates) is moderated depending on the subsequent memory-related activity in two previously identified clusters (see Tab. 2 or Fig. 3). (A) Moderation effect of activation in cluster 1 located around the inferior temporal cortex including fusiform gyrus (positive moderation coefficients). (B) Moderation effect of deactivation in cluster 5 including bilateral cuneus and precuneus as well as posterior cingulate (negative moderation coefficients). The red lines in both panels depict the predicted PACC5 score for a hypothetical individual with an activation 1 SD above the mean, the blue lines for an activation 1 SD below the mean in the respective cluster. The shaded areas represent the 95% confidence intervals. Black dots represent the individual subjects' values for PL and (transformed + residualized) PACC5."

Q1.4 The authors repeatedly signal the validation of the CR measure in their study (repeated in the abstract, discussion, etc), but the generalizability of the CR measure is actually not all that clear to me at present. Consider:

R1.4 We have made significant improvements during revisions of the manuscript which we believe have strengthened the validation procedures regarding the model itself (cross-validation, alternative models, bootstrapping) and its empirical tests, and associations of CR with longitudinal trajectories. We respond to the reviewer's questions one by one below.

Q1.4a The only cross-validation used in model fitting is in determining the optimal number of PCs to reduce the data for the CR estimates. However, this "hyperparameter" doesn't seem to have a big impact on the model. The confidence intervals are huge and the values are barely changing (Fig S5). Would the results change substantially if using e.g. just two PCs? Doing so may actually be to the authors benefit regarding the next point...

R1.4a We thank the reviewer for the helpful feedback regarding the model and validation strategy. In response to the reviewer's suggestion (Q1.4c), we re-calculated the model estimations and correspondingly adjusted the figure (now Fig. S8; see R 1.4c). Thus, in the revised version of the manuscript the confidence intervals decreased significantly. Interestingly, 7 PCs were still found to be superior over 8 and 2 using cross-validation to assess model generalization to unseen data (mean coefficient of determination of 0.3436 for 7 PCs compared to 0.3396 for 8 PCs and 0.3375 for 2 PCs, respectively). We do agree with the reviewer that these differences are not dramatic given the variability in R^2 across the ten independent predictions for each number of PCs. Hence, we also re-evaluated and inspected the model with just two principal components. As expected, the CR pattern displays some differences in terms of the voxelwise CR weights and the specific regions exhibiting significant weights (after bootstrapping). However the results regarding previously identified CR regions such as the fusiform gyrus, angular gyrus, PCC and precuneus were found to be similar, supporting some consistency when using only 2 PCs. The overall correlation of the voxelwise CR weights between 2 and 7 PCs is 0.396 (see new supplemental Fig. S9C, also shown below). The Sørensen-Dice coefficient between the significant regions is 0.489 (see also Fig. S9C). However, for 2 PCs the distribution of the bootstrapped coefficients looks rather bimodal than unimodal normal, which might violate the assumptions of the originally adopted (pooled) bootstrapping approach (Chén et al., 2020)

We therefore revised the bootstrapping procedures to reveal local confidence intervals for the model coefficients (and indicating voxels whose 95% confidence interval does not include 0). All results have been adjusted accordingly and we note that the key CR clusters obtained with the adjusted bootstrapping approach remain similar (Fig. S12 shows the previous results). In the revised analyses, the model with 7 PCs is still the model with the best out-of-sample prediction using CV, and the best cognitive and longitudinal prediction results (see new supplementary material). This suggests that slightly more spatial flexibility is provided by this increased number of basis functions capturing more subtle yet significant patterns in brain activity related to cognitive reserve (slightly better bias variance tradeoff). We believe that despite the increased model complexity, its benefits in explanatory power and cross-validation, justify its inclusion in the main manuscript, with simpler model results using only 2 PCs provided in supplementary material.

We have added the novel results obtained using a multivariate model with only 2 PCs to the supplementary:

*“To study the robustness of the model, we recalculated the findings with a model of significantly reduced complexity. When fitting a multivariate moderation model with only 2 instead of 7 principal components for the functional data (see Fig. S11) and deriving a new CR score based on the coefficients, the corresponding validation models consistently indicated worse predictions of cognitive performance compared to models with the CR score (based on 7 PCs) presented in the main text. One exception was an analysis involving the domain-general cognitive factor in the MRI-only sample. For the cross-sectional analyses in the sample with available PL scores, the R^2 values for the models with the interaction between PL and CR score on cognition were 0.472 for PACC5, 0.486 for the latent memory factor (*f mem*) and 0.416 for the global cognitive factor (*f glob*), in contrast to R^2 values of 0.534, 0.533 and 0.476, respectively, for the CR score based on 7 PCs. In the MRI-only sample with hippocampal atrophy instead of PL scores, the difference in R^2 between 2 versus 7 PCs was less dramatic for PACC5 and *f mem* ($R^2 = 0.425$ for PACC5, 0.442 for *f mem* as compared to 0.441 for PACC5, 0.443 for *f mem* when using 7 PCs). R^2 for *f glob* was slightly higher with 0.363 when using 2 PCs in comparison to 0.353 when using 7 PCs. In the longitudinal model with the three-way interaction of hippocampal atrophy with the CR score and time, the AIC was 2289.2 for a CR score based on 2 PCs as compared to 2267.5 for the CR score based on 7 PCs (reported in the main text), once again indicating a worse model fit with similar interpretations of the three-way interaction. In summary, the validation analyses suggest that a CR score based on a multivariate moderation model in which the functional patterns are described using only 2 principal components instead of 7 is worse at explaining the cognitive performance data, possibly increasing the risk for bias (underfitting). This indicates that two components are likely to be insufficient to adequately represent cognitive reserve patterns in its complexity. On top of that, seven components were superior to any other number of principal components according to the cross-validation.”*

We further acknowledge the reviewer's concern about the susceptibility of the results to the number of PCs (complexity of reserve patterns) in the discussion:

“As the difference in mean cross-validation R^2 was small in comparison to its variability, other choices for the number of principal components (from 2-9) also appear reasonable. Consequently, we would like to note that this model hyperparameter is a non-negligible determinant of the results, susceptible to the bias-variance trade-off. In particular, a model with only two PCs was found to be too restrictive considering the obtained worse model generalization (in cross-validation) and further validation analysis results (see supplementary results and Fig. S11). To support the confidence in the presented results, we provide an illustration of the overlap of CR regions in dependence of the number of PCs and a comparison of models with different numbers of PCs in Fig. S9.”

Despite these differences, the CR scores obtained from the different multivariate models are highly correlated. Overall, the consensus results consistently support the involvement of the DMN, fusiform gyrus, and left hippocampus in cognitive reserve, whereas primary visual areas receive less consistent support. There is also some evidence for the involvement of the right hippocampus in CR. As our model selection favors 7 principal components (updated Fig. S8), we continue to use this model in the main text.

Fig. S9: Significant CR regions across different numbers of principal components. Like in the cross-validation procedure (Fig. S8), the multivariate model has been fitted on the whole dataset using different amounts of principal components, ranging from 2 to 9. Subsequently, the corresponding voxelwise moderation effects (CR weights), voxels with significant CR weights and individual CR scores have been determined. (A) Surface plot of significant regions. (B) Same as A, but as slice views. Color bar indicates how many of the 8 analyses determined a certain voxel as significant contributor to CR. (C) Heat maps comparing the results in dependence of the number of principal components used in the multivariate model based on the correlation of the voxelwise CR weights (lower triangular part) and Sørensen-Dice coefficient (upper triangular part). (D) Same as C, but for individual CR scores obtained from CR weights and subsequent memory coefficients. It is important to note that the CR pattern is multivariate in nature, interpretable as a whole and cluster descriptives are reported for transparency of obtained non-negligible coefficients contributing to the pattern.

Q1.4b In fact, the model weights do not appear to be cross-validated (at least during model fitting). These weights are ultimately what would allow this model to be generalized to new data. Whether this actually occurs later in the manuscript is not entirely clear. If not, there's no real way of testing whether the authors are overfitting to their data here.

R1.4b We thank the reviewer for this important point. While the model weights were indeed not directly cross-validated during model fitting, validation does occur in multiple forms at different stages. We would like to emphasize the importance of the previously discussed cross-validation for selection of the number of principal components. At this stage, the number of model parameters and thus model complexity is determined, reducing the risk of overfitting. Furthermore, bootstrapping of the voxelwise coefficients provides local confidence intervals. After model fitting the parameters are

validated via multiple validation steps of the CR score. First, we use different cross-sectional cognitive test scores, which confirmed a moderation effect of our derived CR score on the association with pathology. Second, we demonstrate a moderation effect of the CR score with hippocampal volumes in an independent validation sample without CSF-based AD biomarker data. Third, in response to the comment of another reviewer, we now also show a longitudinal three-way interaction on cognition between the CR score, (squared) hippocampal atrophy representing pathology and time. We do this both for the whole sample in the main text as well as for both subsamples (the subsample with available PL score and the other without CSF data using hippocampal volumes) in the supplementary. Additionally, we have indicated in the analysis provided in response to the previous question that two principal components seem unable to capture the complexity of a cognitive reserve pattern adequately. In fact, a CR score from a multivariate model with only two PCs achieves worse model fits in almost all of the above-mentioned validation analyses.

Last, we would like to note that our aim was not to build a predictive (AI) model with maximum predictive capabilities for new data, but to rather to propose an explanatory model approach that helps to elucidate the neural mechanisms of cognitive reserve (Shmueli et al., 2010). As one consequence, we consider it a reasonable choice to use the entire available data during model fitting.

To provide more clarity regarding (cross-)validation of model weights, we added the following to the methods: *"While the model was not cross-validated during model fitting, validation occurs in later stages in multiple forms (see section 4.8.3). We additionally note that our aim was not to build a predictive (AI) model with primary focus on predictive capabilities for new data, but to build an explanatory model that helps to elucidate the neural mechanisms of cognitive reserve (see Shmueli⁷² for a comparison)."*

Q1.4c While less important, it should be noted that the authors choose the best model by looking at the mean across folds. This is not correct. Instead, one should look at performance (loss) measured on all out-of-sample data in aggregate. This allows the performance to be estimated across the same amount of data that is observed, while still being all left-out data. It should also be stated what the loss function was? If R-squared, was that literally the square of the r-value (not optimal since it is agnostic to direction) or the actual coefficient of determination?

R1.4c We thank the reviewer for the careful inspection and bringing this mistake to our attention. In response, we have now calculated the performance based on all out-of-sample data in aggregate, to the stark benefit of decreased variability in the coefficients of determination. R-squared refers to the coefficient of determination in this case and has been calculated as $1 - SS_{res}/SS_{total}$, with SS_{res} being the sum of squared residuals and SS_{total} the total sum of squared distances from the mean. We clarified this in the revised text and caption. The figure (shown below), its caption and the section in the supplementary methods have been adjusted accordingly. The latter now reads

"Across the ten folds all data was predicted once based on the remaining 90% for each number of principal components. The coefficient of determination (R^2) between the true and predicted PACC5 values (Box-Cox transformed) was calculated based on the aggregated data, done once per number of principal components. In order to ensure independence of a particular division into folds, this procedure was repeated 10 times with different partitioning of the data into folds. The optimal number of principal components was identified as the corresponding model with the highest mean R^2 value across the 10 predictions."

Fig. S8: **Cross-validation results.** According to Eq. 2, PACC5 was predicted by varying numbers of principal components (eigen-images) in a 10-fold cross-validation procedure that was repeated 10 times with different partitioning of the data (see section 4.7 for details). The boxplots refer to the cross-validation R^2 (coefficient of determination) in PACC5 scores (Box-Cox transformed) in the 10 independent test set predictions. The black line denotes the mean value across the 10 predictions. 7 principal components achieved the best cross-validation results.

Q1.4d P12, L247: “in an analysis encompassing the remaining sample lacking a PL score (due to missing CSF data), a weaker yet significant moderation effect of the CR score on the association between hippocampal volumes and cognitive performance was observed in the form of the latent memory factor ($p = 0.018$, $\beta = 0.131$) and PACC5 ($p = 0.011$, $\beta = 0.145$).” This is perhaps the most convincing and interesting finding in the paper since it may actually indicate model generalization. This is not clear, though, since it is unclear whether this analysis used the parameters directly from the trained model, or whether the model was refit on the new participants using HV instead of PL score? If the former, point 4B above would be mostly satisfied but, if the latter, this analysis is hampered by the same flaw. Either way, this analysis should be visualized. Note that this finding is more interesting (to me) than “validation” with other cognitive tests and cognitive decline in the same subjects, since these factors all tend to correlate pretty strongly (and with task performance) in AD-spectrum samples, so validation would be expected.

R1.4d We much appreciate the helpful comment. The model was not re-fit, but instead the CR scores from the multivariate model ($=A*V*B$; $A = N \times K$ matrix of subsequent memory activity for N participants in K voxels; $V = K \times P$ matrix of K voxel loadings onto P principal components; $B = P \times 1$ matrix/vector with P moderation coefficients from the multivariate model) were used in this analysis. We have included a visualization of this in the revised Fig. 6B (shown below) and adjusted the passage such that it now reads

“Furthermore, the same CR score exhibited a weaker yet significant moderation effect on the association between AD pathology and cognitive performance (see Fig. 6B) for the latent memory factor ($p = 1.17 \cdot 10^{-6}$, $\beta = 0.202 [0.094, 0.310]$, $t = 3.697$, $df = 244$), domain-general factor ($p = 0.020$, $\beta = 0.138 [0.022, 0.254]$, $t = 2.346$, $df = 244$) and PACC5 score ($p = 1.35 \cdot 10^{-6}$, $\beta = 0.271 [0.163, 0.378]$, $t = 4.958$, $df = 240$) in the remaining sample without a PL score (due to missing CSF data). Here, the PL score was replaced by hippocampal atrophy (squared, as the PL score).”

We hope we could address the reviewer’s concern of overfitting with our clarifications and additional analyses.

Fig. 6: CR score is linked to cognitive performance cross-sectionally and longitudinally. (A) The relationship between the PL score and cognitive performance at baseline is moderated by the CR score. Cognitive performance is represented by three different scores: a global cognitive factor score, a memory factor score and the PACC5 score (previously used for identification of the CR-related activity pattern). Cognitive performance was predicted using the respective regression model for an average individual with high (above median; red curve) or low (below median; blue curve) CR score. (B) In the sample without PL score, the CR score moderates the relationship between hippocampal atrophy and cognitive performance. (C) The pathology-dependent differences in longitudinal trajectories of cognitive performance are ameliorated by the baseline CR score. The PACC5 scores at a 5 year follow-up were predicted using the previously described LME (see methods section) fitted on the original longitudinal data for an average individual with high (above median; red line) or low (below median; blue line) CR score and high or low pathology (here represented by high and low hippocampal atrophy corresponding to the 25th and 75th percentile). (D) Predictions of all individual cognitive trajectories. Individuals were categorized as high/low CR based on their above/below average CR scores and as low/high pathology based on their below/above average hippocampal atrophy. Shaded areas in panels A-C denote 95% confidence intervals.

Q1.5 I like the approach to summarize AD pathology with the PL score, though it does come with some limitations. The supplemental data shows the effect of AD pathology on the PL score is quite non-linear. For example, a PL score between 0.4 and 0.6 seems to involve lowish HV and highish p-tau, but normalish $\text{A}\beta_{42:40}$ levels. However, a score between 0.6 and 0.8 indicates abnormal CSF biomarkers but fairly preserved HV. I would not really consider someone in the first category to be “less progressed” than the second category (especially as it pertains to cognition), though they might be more AD-like. In addition, the reason all the results require quadratic function is likely because the relationship between PL score and HV is also quadratic, and HV is probably driving most of the effects in the paper. This seems like a limitation of using t-SNE for this purpose, which really isn’t well designed for what the authors are trying to accomplish. Many such approaches have been described and compared (<https://www.nature.com/articles/s41587-019-0071-9>) that might better suit the authors’ purposes (I wonder if even a simple first eigenvalue approach would do better?).

R1.5 We thank the reviewer for the insightful comment. We selected t-SNE for the purpose of capturing potential non-linearities in AD biomarker relationships, which it obviously did. We do completely agree that the relationships of the biomarkers with the PL score indeed partially show unexpected patterns. However, we would like to note that the observed patterns overall do show considerable face validity. For instance, the higher levels of $\text{A}\beta_{42/40}$ around 0.4 PL with higher p-tau levels and lower hippocampal volumes might represent individuals with different etiologies that are not following a classic AD trajectory.

Following the reviewer's helpful suggestion, we carefully evaluated the effects of multiple alternative methods. In order to retain our assumptions of non-linear patterns in AD pathological load, in addition to PCA we primarily considered methods with the ability to capture those non-linearities. Choosing a linear approach, all of the constructed PL scores independent of the selected method showed very high correlations with each other ($r > 0.9$; see Tab S4, shown below). When re-running the multivariate model with different PL scores obtained from PCA, spectral embedding (SE) or a diffusion pseudotime (DPT) algorithm, the voxelwise moderation (=CR) coefficients were also highly correlated (PCA: 0.907, DPT: 0.878, SE: 0.909) with those from the analysis with the t-SNE-based PL score. The corresponding CR scores show even higher correlations of 0.941 (PCA), 0.929 (DPT) and 0.953 (SE) with the t-SNE-based CR score reported in the main text. Unsurprisingly, the findings were very similar for all CR scores. In both the cross-sectional and longitudinal validation with other cognitive scores in the same sample, the results were the same as with t-SNE-based PL with slightly differing R^2 values and p values for the interaction of the CR score with PL. In summary, based on these additional comparisons the choice of the method for the creation of a PL score does not seem to be critical for the main results, as reasonable representations resemble each other significantly. Since a detailed assessment of the differences between different methods for creating a PL is beyond the scope of this paper, we briefly present the aforementioned alternative PL scores and their results in Figs. S13-S15. Besides our changes to the supplementary, we added the following to the revised discussion: *"In fact, usage of three alternative methods (PCA, spectral embedding, diffusion pseudotime³⁸) produced highly correlated PL scores that in turn yielded very similar CR coefficients (see Figs. S14-S16), indicating that, within reason, the method for obtaining a PL score had only a minor influence on the results."*

	t-SNE	PCA	DPT	SE
t-SNE	1.000	0.930	0.940	0.951
PCA	0.930	1.000	0.965	0.963
DPT	0.940	0.965	1.000	0.983
SE	0.951	0.963	0.983	1.000

Tab. S4: **Correlation between different PL scores.** All PL scores exhibit very high correlations with each other, but show subtle differences in their patterns of non-linearity (see Figs. S14, S15, S16). t-SNE = t-stochastic neighbor embedding, PCA = principal component analysis, DPT = diffusion pseudotime, SE = spectral embedding.

OTHER METHODOLOGICAL QUESTIONS

Q1.6 The reported collinearity between CR activation and basic encoding activation calls to question which is driving the effect in the “validation” analyses. If the authors substitute the encoding parameter (main effect) instead of the CR effect into these models, or puts them both in the model, what happens?

R1.6 We thank the reviewer for this important point. Our original cross-sectional models always included the BAE score (brain additive effect; equivalent of CR score for the main effect of functional activation instead of the interaction effect), PL (squared) as well as an interaction of the CR score with PL² apart from the covariates. If we replace the interaction CR score x PL² by BAE score x PL², the explained variance goes down in all analyses, sometimes substantially. Moreover, the original model does explain more variance in these subsequent analyses:

- CSF/PL subsample: 13.84/11.12/7.91% more variance explained by original model (PACC5/memory factor/global cognitive factor)
- Subsample without CSF/PL (MRI-only): 5.18/2.98/1.49% more variance explained by original model (PACC5/memory factor/global cognitive factor)

When including two additional terms into the models (main effect of CR score and interaction of BAE score with PL²) and comparing these models to the original models in the form of F tests, we obtain the following results:

- CSF/PL subsample: $F(212,2) = 0.294$ for PACC5; $F(216,2) = 1.705$ for the memory factor; $F(216,2) = 0.407$ for the global cognitive factor; all $p > 0.18$. In all models, the main effects of BAE and CR as well as the interaction of CR are significant, but not the interaction of BAE with PL² (smallest $p = 0.083$ for the memory factor, otherwise well above 0.1).
- Subsample without CSF/PL (MRI-only): $F(238,2) = 4.973$, $p = 0.003$ for PACC5; $F(242,2) = 0.521$, $p = 0.109$ for the memory factor; $F(242,2) = 1.042$, $p = 0.595$ for the global cognitive factor. For PACC5 and the memory factor, both interaction effects are statistically significant ($p < 0.05$), but the main effects of BAE and CR are not (all $p > 0.1$). For the global memory factor only the interaction of CR with pathology shows a trend for statistical significance ($p = 0.074$), the main effects and the interaction of BAE with pathology are not statistically significant.
- In all of these "saturated" cross-sectional models, the t value of the interaction of CR with PL is greater than of the interaction of BAE with PL.

In the revised form of the longitudinal models predicting cognitive change, which we had adapted in response to a comment by another reviewer, we included BAE as well as its interaction with time. CR was included, as was its interaction with time, its interaction with (squared) hippocampal atrophy (also ranging from 0 to 1 like the PL score) and the three-way interaction.

When switching the roles of CR and BAE in such a longitudinal model, the AIC of the model increases from 2267.5 to 2314.4, indicating worse model fits. The three-way interaction of BAE with hippocampal volumes and time is not significant ($p = 0.279$). When including both terms with all their two- and three-way interactions with hippocampal volumes and time, the AIC slightly decreases to 2258.8. According to a likelihood ratio test, the full model is slightly better than the original model ($p = 0.005$). In the full model, the three-way interactions of CR and BAE are both significant ($p = 2.41 \cdot 10^{-4}$ and $p = 0.026$, respectively). The two-way interaction with hippocampal atrophy is only statistically significant for CR ($p = 1.98 \cdot 10^{-8}$), for BAE there is only a trend ($p = 0.094$).

In summary, models in which CR is substituted for the main effect of subsequent memory encoding (BAE) consistently fail to achieve similar model predictions. "Saturated" models with both CR and BAE as well as their interactions (with PL^2 for cross-sectional models and all two- and three-way interactions with hippocampal atrophy and time for the longitudinal models) in contrast fit the data slightly better. These models always show stronger interaction effects for the CR score, if any for the BAE score. These results suggest that the identified CR pattern is not just an incidental finding and the observed correlation with activation patterns might reflect biological processes rather than statistical artifacts.

Q1.7 Only some individuals have CSF biomarkers. It would be helpful to know if there was any bias in which participants received both MRI and CSF vs. which received only MRI? What explains this discrepancy and could it be relevant to the study findings?

R1.7 That is an important issue. In terms of demographic variables, there is no observed difference in age at baseline ($p = 0.713$), education ($p = 0.573$), hippocampal volumes ($p = 0.304$), sex ($p = 0.146$) or diagnostic groups ($p = 0.103$) between the sample with both MRI and CSF data ($n_1 = 232$) and the sample with only MRI data ($n_2 = 258$) according to two-sample t tests or Chi square tests, respectively. However, PACC5 scores were significantly lower in the participants which had received both MRI and CSF ($t = -3.473$, $df = 441.16$, $p = 5.67 \cdot 10^{-4}$). The reasons for this remain speculative and multifaceted. Potentially, participants with low cognitive scores were more likely to be motivated to undergo lumbar puncture because they wanted to obtain insights about the cause of their cognitive deficits and potential options on how to treat it or contribute to research against the disease. In any case, as we observed interaction effects of our CR score with hippocampal volumes (squared) in both samples, we assume that this had no effect on the main results and conclusions in this manuscript. To support transparency of the analysis we included a table with subsample descriptives in the revised supplemental material.

Q1.8 The authors should state explicitly somewhere prominent (e.g. Results) that their QC procedure disproportionately removed cognitively impaired individuals — only 14% of subjects were removed, but this included over half of AD patients and almost a third of MCI patients.

R1.8 We thank the reviewer for this helpful remark. We can reassure the reviewer that, since the original sample had consisted of 558 subjects, including 83 MCIs and 33 ADDs, the proportions of MCIs and ADDs removed were "only" 22.9% and 36.4%, respectively. As this is still a substantial difference, we note it in the first section of the results to make this transparent:

"We note that of the 58 participants (10.4% of the original sample) that had been excluded from the analyses, the proportion of aMCI (n = 19; 22.9% of the 83 in the original sample) and ADD (n = 12; 36.4% of the 33 in the original sample) was disproportionately higher due to their greater movement during fMRI, response bias etc. (see supplementary methods)"

Q1.9 It is confusing that the authors use PCs to summarize the CR data, but then later break the results into clusters. Are these clusters distinct from the PCs?

R1.9 We thank the reviewer for pointing this important issue out. We completely agree that the identified CR-related activity pattern using a multivariate model is a spatially distributed pattern (described using a linear basis obtained from PCA) that should be rather interpreted as an entity. Therefore, single clusters should not be interpreted as reflecting region-specific effects (e.g. of reserve) but as contributions to the whole pattern. Hence, the message we aim to convey is a transparent representation of the obtained CR coefficients (reflecting the local contributions to the pattern) as presented in Fig. 2B. We further make an attempt to assess the confidence/stability of the observed patterns via bootstrapping similar to previous applications of multivariate latent variable approaches in the field (e.g. Partial Least Squares (PLS) for brain-behavioral analysis, McIntosh & Lobaugh, *Neuroimage*, 2004, Ziegler et al., *Neuroimage*, 2013). Similar to other applications of multivariate latent variable models for analysis (rather than prediction), the number of latent variables (here PCs) is a hyperparameter that enables more or less complex (brain-behavioral- or reserve-) patterns to be identified, respectively. The motivation to report details on spatial clusters that we observed for the CR pattern coefficients after applying some reasonable threshold (from bootstrapped local confidences) is transparency. This follows similar procedures from above mentioned PLS analysis where bootstrapping enables ignoring negligible pattern coefficients (due to high noise or variability). To avoid overinterpretation as region-specific effects we now better emphasize the multivariate nature of patterns more clearly in the caption and text of the revised version of the manuscript.

More generally the PCs are the spatial basis functions to describe any generic pattern in a linear combination and not necessarily identical to clusters. Interestingly, single PCs (i.e. weight-space directions of maximum across-subject variation of activity) do not closely align with the identified CR pattern. That being said, the CR-related activity pattern is most similar to PCs 4, 7, 5 and 2. The correlations of the voxelwise CR weights with the principal component loadings is 0.568, -0.497, 0.469 and 0.395 for 4, 7, 5 and 2 PCs, respectively. PCs 6, 1 and 3 contribute weaker to the CR pattern. Their correlations are 0.209, -0.172 and -0.105, respectively. These correlations also indicate why a pattern described using only 2 PCs seems to be insufficient to capture the complexity of CR-related activation differences.

Q1.10 In Figure 5 — why aren't there confidence intervals? In Figure 6B — why do the number of timepoints only go up to 5, and why does it seem like everyone has 5 timepoints (the methods stated up to 6 timepoints but not everyone had all timepoints)? Why does it seem like only two points are visualized?

R1.10 We very much appreciate the insightful observation of these important details. The lines in Fig. 5 are calculated for a hypothetical subject with brain activity equal to the mean activity \pm 1SD in every voxel of the cluster, respectively. As also mentioned in response R1.4a, we adapted the voxel inference approach to a more traditional approach, as for fewer principal components the distribution was not normal as expected/required. Additionally, this allowed us to obtain 95% confidence intervals for the CR coefficients of every voxel that in turn could be used to quantify the uncertainty of predictions based on the activity of the voxels. Hence, as requested by the reviewer, in the revised version of the manuscript we included confidence intervals in Fig. 5.

We apologize for the potential confusion that the former Fig. 6B might have caused. As we have now adjusted the longitudinal model in response to the comment of another reviewer, the results are now slightly different and we decided to modify the figures slightly. Fig. 6C now essentially shows the three-way interaction of CR with pathology and time according to the revised model in a similar fashion as for the cross-sectional models in Figs. 6A and B. Revised Fig 6D is similar to the previous Fig. 6B. It now includes all participants. Please note, however, that the predicted individual cognitive trajectories of all participants according to a fitted LME are displayed (i.e. model predictions), not their actual data points. To avoid potential confusion with observed data, we removed points (predicted y for fixed timepoints) and show only predicted trajectory curves in a continuous fashion (see Fig. 6 in R1.4d).

Q1.11 A FEW LINGERING MINOR POINTS

R1.11 We thank the reviewer for these helpful observations and suggested corrections. We have implemented those to improve the manuscript.

Q1.11a P19 line 482 “Participants were classified as MCI when displaying an age-, sex-, and education-adjusted performance below -1.5 SD on the delayed recall trial of the CERAD word-list episodic memory tests” — would it be fair to call these subjects amnesic MCI?

R1.11a Correct. We have renamed them accordingly.

Q1.11b For CSF analysis, while full details are not necessary, please at least record which approach/lab/company processed the data

R1.11b We have included the following sentence to provide information about the CSF processing in the revised manuscript methods section:

"AD biomarkers were determined using commercially available kits according to vendor specifications: V-PLEX A β Peptide Panel 1 (6E10) Kit (K15200E) and V-PLEX Human Total Tau Kit (K151LAE) (Mesoscale Diagnostics LLC, Rockville, USA), and Innotest Phospho-Tau(181P) (81581; Fujirebio Germany GmbH, Hannover, Germany)."

Q1.11c Please visualize the CR vs education score. Beta and p-values alone don't tell us much, it helps to see the data and there is plenty of room in the supplementary.

R1.11c We are thankful for the helpful suggestion. We included it in Fig S20A, which is shown below.

Q1.11d The sentence on P15 that starts at Line 312 seems completely out of nowhere. It seems to cite a poster or something, and is probably not the best example of the point the authors are making considering other published work out there.

R1.11d We thank the reviewer for pointing it out. We removed the sentence and the statement about genetic mechanisms, as it remains rather speculative.

Q1.11e Fig 2A images would be easier to ascertain if projected to a cortical surface, as in 3B. But please in these (and those in 3B) include a medial view. There are now ample tools for quickly and easily plotting volumetric data onto surfaces

R1.11e That is a helpful suggestion. We have included surface plots in revised Figs. 2 and 3 to improve visualization of our findings. The revised figures are shown below.

Fig. 2

Fig. 3

Reviewer #2 (Remarks to the Author):

Q2.0 This is an excellent, very well-written and scientifically rigorous paper investigating the neural implementation of Cognitive Reserve (CR) across the Alzheimer's clinical spectrum utilizing task-related fMRI and A/T/N biomarkers. The work has several strengths - I'd emphasize its robust conceptual framework, the longitudinal validation of the approach, and the meticulous justification of each methodological choice.

I only have a few questions and comments:

R2.0 We appreciate the enthusiastic and positive assessment of our study and strengths of the manuscript.

Q2.1 The neural implementation of CR may vary across disease stages. While I understand that this is, to some extent, accounted for as PL is continuously measured across the disease, it would be interesting to discern whether the implementation of CR remains consistent between cognitively normal, SCD, MCI or AD patients or - if sample size does not allow - between non-demented and demented individuals. Could the authors calculate and project the moderations coefficients (i.e., CR pattern) within those groups?

R2.1 We very much appreciate the stimulating comment and interesting scientific question about reserve processes as a function of the actual progress towards AD. As the reviewer already alluded to, sample size only allowed a split into cognitively impaired (CI; MCI + ADD) and cognitively unimpaired (CU; HC, ADR, SCD) participants. We re-ran the model separately for these two groups and present the results in Fig. S17 (see below). For the CUs the new results were found to be similar to the obtained results in the whole sample (correlation of 0.891 with CR coefficients from the whole sample). Interestingly, for the CIs the coefficients look very different (correlation of 0.150 with CR coefficients in the whole sample). While this is a very interesting finding, we would advise against overinterpretation. The sample size of CIs was only 44 compared to 184 CUs, while the model has a considerable amount of parameters. Taken together with the considerable variability of functional activation data this increases the chance of overfitting in these smaller samples. As expected, these two groups also differ in several important aspects. The CIs are older (72.52 vs 68.95 years, $p = 2.30 \times 10^{-5}$), have higher pathological load (PL scores, 0.637 vs 0.359, $p = 2.52 \times 10^{-8}$), show a lower cognitive performance (PACC5 scores, -1.76 vs -0.02, $p = 2.40 \times 10^{-12}$) and are less educated (13.36 vs 14.82 years, $p = 5.02 \times 10^{-4}$). The latter might be another indication that brain activity patterns underlying CR may not only differ in CIs, but moreover that CIs might simply have lower CR that made them more susceptible to cognitive deficits in the first place. In consequence, they might not be a good model for examining CR. Dedicated samples of cognitively impaired participants and larger sample sizes might be better suited to examine the neural mechanisms of CR in this or other special subgroups.

We added the following sentence to the discussion:

"Furthermore, the current study focuses on an approach that assumes sample-level identification of reserve patterns, while ignoring the possibility that CR might actually be implemented differently across different subpopulations or disease stages. For instance, CR-related activity patterns might differ between early and late disease stages⁵⁹ or males and females (see Figs. S17 and S18)."

and note the cautions in the caption of Fig. S17, which is shown below.

Fig S17: Separate CR coefficients for CIs and CUs. (A) Results of the multivariate model when applied only to cognitively impaired people (aMCI, ADD; $N = 44$; $r = 0.150$ with CR coefficients in main text). (B) Results of the multivariate model when applying it to only cognitively unimpaired people (HC, ADR, SCD; $N = 184$; $r = 0.891$ with CR coefficients in main text). Please note that the substantial difference in sample sizes and many variables between the two groups. The CIs are older (72.52 vs 68.95 years, $p = 2.30 \cdot 10^{-5}$), have higher PL scores (0.637 vs 0.359, $p = 2.52 \cdot 10^{-8}$), lower PACC5 scores (-1.76 vs -0.02, $p = 2.40 \cdot 10^{-12}$) and are less educated (13.36 vs 14.82 years, $p = 5.02 \cdot 10^{-4}$). The latter might be another indication that CR not only may work differently in CIs, but CIs might also simply have lower CR that made them more susceptible to cognitive deficits in the first place. In consequence, they might not be a good model for examining CR. Dedicated samples of cognitively impaired participants might be better suited to examine the neural mechanisms of CR in this special subgroup.

Q2.2 Similarly, given the increasing interest and acknowledgment of sex/gender differences in reserve and resilience, it'd be interesting to know whether the CR pattern is different in males and females. This could be presented a supplemental material.

R2.2 In response to the reviewer's request, the reserve analysis was re-run separately for both sexes and the results are presented in Fig. S18 of the revised supplemental material (shown below). Within male participants the CR coefficients strongly align with those presented in the main results ($r = 0.879$), but less so for the females ($r = 0.282$). This is an interesting finding acknowledged in the revised discussion of the manuscript, as stated above in our response R2.1.

However, for reasons of potential biases in less powered analyses using smaller subsamples, we do believe larger future studies might be better suited to focus on sex/gender effects.

Fig S18: **Separate CR coefficients for females and males.** (A) Results of the multivariate model when applying it to only female participants (N = 114; $r = 0.282$ with CR coefficients in main text). (B) Results of the multivariate model when applying it to male participants (N = 114; $r = 0.879$ with CR coefficients in the main text).

Q2.3 If my understanding is accurate (line 598) all the analyses were restricted to the task-related fMRI activation mask. If so, the potential engagement of additional areas remains unexplored. Consequently, the claims regarding cognitive reserve being implemented as a sustained recruitment of core cognitive circuits (as opposed to the activation of alternative regions) may need qualification. Could you please clarify this aspect?

R2.3 We thank the reviewer for this interesting point. We refer to our above response R1.2, where we discuss several aspects of this finding and offer additional analyses.

Q2.4 I was surprised of the choice of creating a PL score that incorporated both Alzheimer's disease pathologies and hippocampal volume. I understand the rationale behind representing A/T/N. However, hippocampus is involved in memory encoding and higher hippocampal volume may potentially serve as a measure of both disease progression (neurodegeneration or atrophy) and of brain reserve as it may be influenced by the exposure to protective factors (e.g. exercise). I wonder whether this methodological decision has impacted the results and explains some of the discordant results within the hippocampus. Could the authors clarify whether the results are similar when only amyloid and tau are considered?

R2.4 We thank the reviewer for this insightful comment and agree that a cross-sectional volumetric measure could reflect neurodegeneration or brain reserve (for any given region), which remains a limitation. However, we note that the results are comparably similar and consistent with our current findings when removing hippocampal volumes from the PL score. First, these alternative PL scores are highly correlated with the original ATN-based PL score ($r = 0.845$ when using t-SNE, $r = 0.865$ when using PCA to create the AT-based PL score). Second, the CR coefficients obtained from multivariate reserve models with the corresponding PL scores are also strongly associated with the ones provided in the main results of our manuscript ($r = 0.961$ for t-SNE, $r = 0.866$ for PCA). This can be also further seen in the now included Fig. S16, shown below.

Fig S16: **PL score from $A\beta_{42:40}$ and p-tau only.** Here we present an alternative PL score based only on $A\beta_{42:40}$ and p-tau, excluding hippocampal volumes. (A) Relationship of a t-SNE-based AT PL score with perplexity 25 ($r = 0.845$ with original PL score) to $A\beta_{42:40}$ and p-tau. (B) Voxelwise moderation coefficients (CR) for multivariate model with t-SNE-based AT PL score ($r = 0.961$ with CR coefficients in main text), normalized by the highest absolute coefficient. (C) Relationship of a PCA-based AT PL score ($r = 0.865$ with original PL score) to $A\beta_{42:40}$ and p-tau. (D) Voxelwise moderation coefficients (CR) for multivariate model with PCA-based AT PL score ($r = 0.866$ with CR coefficients in main text), normalized by the highest absolute coefficient. Color scale below panel C applies to the PL score of both panels A and C. Color scale below panel D applies to the CR coefficients in both panels B and D.

Q2.5 The authors discuss various potential neural mechanisms underlying the identified CR activation pattern, including vascular supply, clearance, inflammation, and compensation by non-affected neural populations within the cognitive reserve circuit. Examining the structural integrity, such as brain volumes in the CR-related regions, could provide insights into this latter aspect.

R2.5 We appreciate the helpful suggestion. The structural integrity in CR-related regions, represented by mean gray matter volumes (derived from SPM segmentation) from all voxels with significant contributions to CR was weakly correlated with the CR score ($r = 0.112$, $p = 0.013$). This indeed suggests

a small contribution of retained structure to CR. However, we also made sure that our results of a CR-related activity pattern were independent of the observed morphological differences. We report these new findings in the supplementary results section:

“The CR score was further weakly positively correlated with total GM volumes in the regions with significant contributions to CR ($p = 0.013$, $r = 0.112$ [0.024,0.199], $df = 487$) and mean cortical thickness ($p = 0.023$, $r = 0.103$ [0.015,0.190], $df = 487$). Yet, it should be noted that accounting for these structural differences as covariates had essentially no effect on the observed results, neither in the multivariate moderation model nor the validation analyses of the CR score (see Fig. S19) and Tab. S1. Taken together, although AD pathology indices, tissue volumes and white matter lesions were slightly associated with individual CR score differences, the pattern of CR-related brain areas and its predictive value for memory performance was not mediated by atrophy alone or network connectivity at rest.”

Fig. S19: **CR coefficients when including morphometric covariates.** (A) Results of the multivariate model when including mean cortical thickness (from Freesurfer) as an additional covariate ($r = 0.999$ with CR coefficients in main text). (B) Results of the multivariate model when including mean GM volumes (from SPM segmentation) of the voxels with significant CR contributions (according to the main model) as an additional covariate ($r = 0.995$ with CR coefficients in the main text).

Q2.6 Minor:

Q2.6a The authors indicate the existence of 51 first-degree relatives of Alzheimer's Disease (AD) patients. Could you provide clarification on whether all participants were assessed for a history of sporadic AD, and if so, how was this assessed?

R2.6a The assessment relied on self-reports by the participants of the DELCODE study. All participants who were recruited via advertisement had been asked if they had a person in their family with diagnosis of Alzheimer's dementia. Moreover, it is important to note that if it was not clear, or other types of dementia were reported (for relatives), then participants were included in the study as cognitively normal.

Q2.6b Line 176 contains a typo it should read “as well as” instead of “as well”

R2.6b We thank the reviewer for spotting this. The typo is corrected.

Q2.6c Lines 204-206: The authors may want to check the work by Elman and colleagues doi: 10.1038/nn.3806

R2.6c We thank the reviewer very much for bringing the work of Elman et al. back to our attention. Importantly, their study discovered a cluster in medial parietal cortex, in which deactivation was parametrically modulated by memory encoding (detail in this case) across the whole sample, but in which young adults showed no such parametrically modulated deactivation, similar to our principle of discordant regions. We discuss this finding in the discussion:

"Similar to our findings regarding discordant regions, previous reports have identified comparable phenomena. For instance, Elman et al.⁴⁹ discovered a cluster in medial parietal cortex in which deactivation was parametrically modulated by the level of memory encoding detail across the entire sample of young and older adults with and without A β . In this cluster, greater deactivation was associated with higher detail recall. However, young adults exhibited no such deactivation, resembling our observations of discordant regions."

Q2.6d Could you please confirm – and provide clarification – that the dimensional reduction for PL calculation, considers the distinct directions of the variables where CSF A β and hippocampus volume (less is worse), and CSF tau (more is worse)?

R2.6d Yes, the pairwise correlations with the PL score were -0.862 for A β _{42:40}, 0.670 for p tau and -0.684 for the TIV-corrected hippocampal volumes, i.e. more p-tau resulting in a higher pathological load, whereas a lower A β ratio and lower hippocampal volumes was associated with more pathological load. This information is presented in the supplementary (section 5.2.1).

Q2.6e Line 366: the reference to the results by Franzmeier et al. identifying the left prefrontal cortex (when discussing findings on the fusiform, temporal and occipital regions) seem to lack specific discussion. Were the authors surprised by the absence of findings in the left prefrontal cortex?

R2.6e We apologize for any confusion that our wording might have caused. Franzmeier et al. discovered that the global connectivity of the left frontal cortex (seed in -42,6,28 in MNI coordinates) moderated (attenuated) the association between precuneus hypometabolism measured by FDG-PET as measure of AD pathology and lower memory performance, in line with the proposed moderation framework for cognitive reserve. While we did not examine connectivity measures in our study, we also found a cluster in the left frontal cortex whose activity during successful memory encoding (as part of our whole multivariate pattern) attenuated the association between AD pathological load and worse memory performance. This cluster is denoted cluster 4 in the revised Tab. 2 and in fact contains the seed used in Franzmeier's study. Hence, the results are in good alignment in our opinion.

We revised the section in the discussion to *"Further, some frontal regions have been proposed to play a role in cognitive reserve. For instance, Franzmeier et al.⁷ discovered that global connectivity of the left frontal cortex attenuated the relationship of precuneus FDG-PET hypometabolism (as proxy for AD severity) on lower memory performance in amyloid-positive individuals with aMCI. The left frontal cortex also showed positive contributions to CR in our study (cluster 4 in Tab 2.; see also Fig. 3)."* in an attempt to improve clarity.

Reviewer #3 (Remarks to the Author):

Q3.0 The present study examined whether patterns of brain activity during memory formation may constitute a neural basis of cognitive reserve (CR) as defined by recent research criteria. They found that a multivariate pattern of activity related to the subsequent memory (SM) effect moderated the cross-sectional association between a multivariate pathology score and cognitive performance. They then generated subject-specific values reflecting the degree to which an individual's encoding activity resembled this pattern and found that higher similarity was associated with less longitudinal decline. Higher similarity scores were also correlated with education, a commonly used measure of cognitive reserve.

Overall, this is a well-written manuscript with the potential to contribute novel insight into neural mechanisms that underlie resilience to AD-related processes. The authors present strong conceptual grounding to motivate their study, using a definition of CR that focuses on how a given measure moderates the relationship between pathology and cognition. The multivariate methods proposed here may represent a useful approach to studying CR in other contexts. There are however some key conceptual and methodological questions that remain, and the manuscript would benefit from their clarification. For example, results from longitudinal analysis can provide strong support for CR, but the model as currently specified may be just as likely to detect differences in disease progression/staging as differences in CR. These issues are detailed below, as well as some minor suggestions.

R3.0 We thank the reviewer for the positive evaluation of our study.

Major Comments:

Q3.1 In looking at the variance explained by the principal components, it seems that the first 2 PCs may be sufficient to capture the bulk of the variance. Does an analysis using this more parsimonious basis set perform similarly ?

R3.1 We thank the reviewer for the important question. A similar question regarding model selection and appropriate complexity was discussed in above response R1.4a to another reviewer. In order to optimize the tradeoff of model complexity (in terms of numbers of PCs or basis functions) and fit we opted for the assessment of a test error (in 10-fold cross-validation) as a metric for model selection, since it is a direct metric of how well the model generalizes to new, unseen data. We assumed that a better characterization of the underlying reserve pattern (with a-priori unknown complexity, e.g. regional differences) was assumed to result also in better generalization. This choice of procedure and metric is motivated in machine learning textbooks that propose test error as one suitable metric to find good compromises of the bias and variance tradeoff (Bishop, 2006, Murphy, 2022). The model complexity that minimized this empirical test error was considered optimal, resulting in a 7 PCs as (spatial) basis set for the reserve patterns in our sample. As the reviewer suggested, one might *a-priori* choose a reduced model complexity. This might in turn increase the risk of bias by spatially simplifying the reserve patterns, for potentially less variable model parameters (e.g. due to noise). In order to address the reviewer's concern we therefore repeated the reserve model specification and analysis using 2 instead of 7 principal components. Given this inherent constraint on the complexity of estimable CR patterns, we expectedly observed some differences in terms of the local CR weights (moderation coefficients) and the specific regions showing significant contributions to the pattern.

However, it is also important to state that the findings with regards to the previously identified CR regions (using a more complex model) in the fusiform gyrus, angular gyrus, PCC and precuneus were found to be similar and therefore supporting some degree of consistency across both models. More empirically, the correlation of the voxelwise CR weights obtained from models with 2 and 7 PCs (as basis sets) is 0.396 (see new Fig. S9C, shown below). The Sørensen-Dice coefficient between the regions significantly contributing to the reserve pattern was found to be 0.489 (also Fig. S9C).

In terms of the subsequent validation analyses of the CR score, the corresponding models with the new CR score (using a simpler basis set) almost exclusively achieved worse out-of-sample predictive accuracy (in cross-validation) as compared to before. In addition, we added new analyses comparing the predictive value of CR scores (from both above models) for longitudinal data using a triple interaction of (squared) hippocampal atrophy with the CR score and time (we adapted this model in responses to the helpful suggestions in the later comment Q3.5). The AIC (model comparison index) was found to be 2289.2 for a CR score based on 2 PCs as compared to 2267.5 for the CR score reported in the revised main results (using 7 PCs), once again indicating a worse model performance (while retaining similar interpretations for the triple interaction).

In summary, in addition to model selection procedures (based on rigorous textbook principles) indicating better predictions of cognitive outcomes when using a richer reserve pattern, the additional validation analyses also suggest that a simplified 2-dimensional spatial (PCA) basis instead of a 7-dimensional one is worse at explaining the individual differences of cognitive performance as well as trajectories over time, increasing the risk for oversimplification and biased results.

We mention the results for the multivariate model with 2 PCs in the supplementary material:

“To study the robustness of the model, we recalculated the findings with a model of significantly reduced complexity. When fitting a multivariate moderation model with only 2 instead of 7 principal components for the functional data (see Fig. S11) and deriving a new CR score based on the coefficients, the corresponding validation models consistently indicated worse predictions of cognitive performance compared to models with the CR score (based on 7 PCs) presented in the main text. One exception was an analysis involving the domain-general cognitive factor in the MRI-only sample. For the cross-sectional analyses in the sample with available PL scores, the R^2 values for the models with the interaction between PL and CR score on cognition were 0.472 for PACC5, 0.486 for the latent memory factor (f_{mem}) and 0.416 for the global cognitive factor (f_{glob}), in contrast to R^2 values of 0.534, 0.533 and 0.476, respectively, for the CR score based on 7 PCs. In the MRI-only sample with hippocampal atrophy instead of PL scores, the difference in R^2 between 2 versus 7 PCs was less dramatic for PACC5 and f_{mem} ($R^2 = 0.425$ for PACC5, 0.442 for f_{mem} as compared to 0.441 for PACC5, 0.443 for f_{mem} when using 7 PCs). R^2 for f_{glob} was slightly higher with 0.363 when using 2 PCs in comparison to 0.353 when using 7 PCs. In the longitudinal model with the three-way interaction of hippocampal atrophy with the CR score and time, the AIC was 2289.2 for a CR score based on 2 PCs as compared to 2267.5 for the CR score based on 7 PCs (reported in the main text), once again indicating a worse model fit with similar interpretations or the three-way interaction. In summary, the validation analyses suggest that a CR score based on a multivariate moderation model in which the functional patterns are described using only 2 principal components instead of 7 is worse at explaining the cognitive performance data, possibly increasing the risk for bias (underfitting). This indicates that two components are likely to be insufficient to adequately represent cognitive reserve patterns in its complexity. On top of that, seven components were superior to any other number of principal components according to the cross-validation.”

We further acknowledge the susceptibility of the results to the number of PCs in the discussion: *“As the difference in mean cross-validation R^2 was small in comparison to its variability, other choices for the number of principal components (from 2-9) also appear reasonable. Consequently, we would like to note that this model hyperparameter is a non-negligible determinant of the results, susceptible to the bias-variance trade-off. In particular, a model with only two PCs was found to be too restrictive considering the obtained worse model generalization (in cross-validation) and further validation analysis results (see supplementary results and Fig. S11). To support the confidence in the presented results, we provide an illustration of the overlap of CR regions in dependence of the number of PCs and a comparison of models with different numbers of PCs in Fig. S9.”*

Despite these differences, it is worth mentioning that the CR scores obtained from the different multivariate models are highly correlated. Overall, the consensus results seem to support the involvement of the DMN, fusiform gyrus and left hippocampus in cognitive reserve patterns. In contrast, primary visual areas receive less support. There is also some evidence for the involvement of the right hippocampus in CR. As provided model selection results (and validation analyses) did favor patterns using 7 principal components (updated Fig. S8, shown below), we would argue for abiding by this model.

Fig. S9: Significant CR regions across different numbers of principal components. Like in the cross-validation procedure (Fig. S8), the multivariate model has been fitted on the whole dataset using different amounts of principal components, ranging from 2 to 9. Subsequently, the corresponding voxelwise moderation effects (CR weights), voxels with significant CR weights and individual CR scores have been determined. (A) Surface plot of significant regions. (B) Same as A, but as slice views. Color bar indicates how many of the 8 analyses determined

a certain voxel as significant contributor to CR. (C) Heat maps comparing the results in dependence of the number of principal components used in the multivariate model based on the correlation of the voxelwise CR weights (lower triangular part) and Sørensen-Dice coefficient (upper triangular part). (D) Same as C, but for individual CR scores obtained from CR weights and subsequent memory coefficients. It is important to note that the CR pattern is multivariate in nature, interpretable as a whole and cluster descriptives are reported for transparency of obtained non-negligible coefficients contributing to the pattern.

Fig. S11: **CR pattern in multivariate model with only 2 principal components.** Results of the same multivariate model using 2 instead of 7 PCs. (A) Voxelwise moderation coefficients (CR), normalized by the highest absolute coefficient. (B) Regions with significant CR coefficients as indicated by bootstrapping. It is important to note that the CR pattern is multivariate in nature, interpretable as a whole and cluster descriptives are reported for transparency of obtained non-negligible coefficients contributing to the pattern

Fig. S8: **Cross-validation results.** According to Eq. 2, PACC5 was predicted by varying numbers of principal components (eigen-images) in a 10-fold cross-validation procedure that was repeated 10 times with different partitioning of the data (see section 4.7 for details). The boxplots refer to the cross-validation R² (coefficient of

determination) in PACC5 scores (Box-Cox transformed) in the 10 independent test set predictions. The black line denotes the mean value across the 10 predictions. 7 principal components achieved the best cross-validation results.

Q3.2 Related to the above, does a similarity score for each individual relative to the group SM contrast (or an SM contrast within cognitively normal only) produce similar results? If so, it could simplify interpretations (e.g., higher/lower activity may be easier for readers to understand than voxel or subject loadings summed across multiple PCs). There were discordant regions between the CR-related map and SM map, indicating the maps are not completely redundant, but it is unclear how much of an impact these discrepancies would have on the results.

R3.2 We thank the reviewer for the thoughtful and inspiring suggestion. In response to the reviewer's comment, we defined and calculated a new similarity score as a t-weighted sum of the voxelwise individual activations. The t values were obtained from a t-contrast of the parametric subsequent memory regressor in the sample of only cognitively normal individuals. Such a score is substantially correlated with the CR score ($r = 0.556$). In the validation analyses, the results are similar, although the models with a similarity score instead of the CR score explain less variance in the cognitive outcomes in the sample with AD biomarker data. More specifically, for the cross-sectional analyses in the sample with PL scores, the R^2 values when using the t-weighted sum score were 0.460 for PACC5, 0.503 for the latent memory factor and 0.422 for the global cognitive factor (in comparison to higher R^2 values of 0.534, 0.533 and 0.476, respectively, for the CR score). In the MRI-only sample with hippocampal volumes instead of PL scores, R^2 values were also slightly lower ($R^2 = 0.411$ for PACC5, $R^2 = 0.426$ for f_mem, $R^2 = 0.351$ for f_glob for the t-weighted sum score as compared to $R^2 = 0.441$, 0.443, 0.353 for the CR score) with similar p values. For the domain-general cognitive score, the p value of the interaction of the similarity score with hippocampal atrophy was not significant ($p = 0.277$). Furthermore, in the longitudinal model the AIC was higher, i.e. worse, when using the t-weighted score (2304.9 instead of 2267.5).

While this approach seems very noteworthy, we believe that the proposed moderation approach offers certain conceptual and empirical advantages. First, the similarity score can only support general statements such as "it is beneficial for cognitive performance if one's successful encoding activity resembles that of cognitively normal individuals", whereas the (multivariate) moderation approach allows a more specific investigation and identification of activity patterns of cognitive reserve, e.g. allowing us to identify discordant regions. Furthermore, the similarity score likely only works so well because the parametric subsequent memory contrast inherently represents memory performance differences to some degree. In contrast, the moderation approach could be applied to all kinds of fMRI task data. It could even be applied to other forms of imaging data such as resting state fMRI or structural imaging features. Third, the consensus framework by Stern et al. strongly suggests to perform moderation analysis for the investigation of cognitive reserve, which we here follow closely (thanks to the reviewer's comment now also more closely in the longitudinal setting).

We added the results to the supplementary:

"An alternative approach to creating a CR score, which is not based on the consensus framework, was to quantify the similarity of the individuals' brain activity during successful memory encoding with the average activity of the cognitively normal individuals (similar to previous approaches like the FADE⁵⁵ or FADE-SAME scores²⁶). Hence, a similarity score was calculated as a t-weighted sum of the individuals' parametric SM contrast with the group-level t contrast of the cognitively normal

*individuals. Indeed, the similarity score was correlated substantially with the CR score ($r = 0.556$). The results in the validation analyses were also similar, although the models with a similarity score instead of the CR score explained slightly less variance in the cognitive outcomes. For the cross-sectional analyses in the sample with PL scores, the R^2 values were 0.460 for PACC5, 0.503 for the latent memory factor and 0.422 for the global cognitive factor when using the t-weighted sum score (0.534, 0.533 and 0.476, respectively, for the CR score). In the MRI-only sample with hippocampal atrophy instead of PL scores, R^2 values were lower for the t-weighted sum score ($R^2 = 0.411$ for PACC5, 0.426 for *f mem*, 0.351 for *f glob* as compared to $R^2 = 0.441$, 0.443, 0.353, respectively). For the domain-general cognitive score, the p value of the interaction of the similarity score with hippocampal atrophy was not significant ($p = 0.277$). In the longitudinal model, the AIC was higher, i.e. worse, when using the t-weighted sum score (2304.9 compared to 2267.5 for the original CR score). While this is noteworthy, the disadvantages of the similarity score compared to the proposed explicit approach are briefly addressed in the discussion of the main text.”*

We also included a small paragraph in the discussion:

“Of note, an ad hoc definition of a similarity score representing an individual’s activity similarity to group-level successful encoding activity in cognitively normal individuals (similar to the idea of the FADE⁵⁵ or FADE-SAME score²⁶) also shows considerable moderation effects of pathology on cognitive performance in our sample (see supplemental material). Yet, our proposed approach offers some advantages. First, it explicitly assumes and tests the moderation as specified in the recent consensus framework on the level of brain activity patterns using task fMRI. Second, this multivariate moderation approach allows for more specific findings, such as the identification of above presented discordant regions. Third, the moderation approach can be extended to other indicators of pathology, memory-independent contrasts, multi-task data or other forms of reserve-related imaging features such as resting-state fMRI.”

Q3.3 Higher similarity scores are interpreted as individuals with high CR, but an alternative is that lower similarity scores reflect individuals with more progressed disease. Although the PL score controls for hippocampal atrophy, it does not rule out individual differences in overall atrophy. As an example of this alternative explanation, the panels of Figure 6B could be reinterpreted as showing quartiles of disease staging rather than CR. Individuals in quartile 4 may be considered to have lower resilience, or they may just be further along in the disease process, with more widespread atrophy causing disruptions to encoding activity and lower cognitive performance. The possibility that measures of CR are recapitulating measures of disease progression is in no way specific to this study – it is difficult to rule out in many studies of CR. However, it should be addressed, whether through further analysis or in the discussion.

R3.3 We thank the reviewer for the insightful comment. We first like to point out that by aiming to implement the previously suggested reserve analysis framework (via moderation) a core assumption is a conceptual distinction of any potential CR-variable and the pathology variable. The causal structure of these reserve mechanisms could be interpreted as follows. Some kind of brain pathology (e.g. atrophy) in older participants causes brain networks to suboptimally function, resulting in a decline in their cognitive performance (ideally longitudinally). We understand the reserve variable as an additional context property (e.g. functional or even structural) of the very same brains, that means individual variability which might have originated due to other factors such as genes, or previous

environmental experiences like education or lifestyle. One might here assume that these latter causes are rather independent from the ones driving the pathology in the first place. This assumption then would translate in the statistical moderation approach in that pathology and reserve variables interact (suggesting non-linear mechanisms) such that the latter moderates the effects of the former. Under these assumptions the reserve variable could be further parametrized as in our proposed approach. We do agree with the reviewer that the ad hoc choice of a pathology variable and distinct reserve variables (e.g. from different imaging modalities as done here) might appear somewhat arbitrary. However, we would argue that the particular choices (ATN-based pathology and fMRI activation patterns as reserve) are aligned with the reserve framework's suggestions. The rather implicit assumption of independence of primary variables might indeed empirically be violated (e.g. existence of some genes that partially affect both, pathology and reserve) resulting in multicollinearity of these predictors. This, however, is an empirical question and downstream correlations of proxies of processes of interest are a very common challenge in many well published observational studies. In this framework we understand the individual stage being equivalent to the pathology variable and that the subsequent reserve moderation analysis itself is a group-level analysis which relies on individual variability in both the pathology status (or stage) as well as the reserve variable. Consequently by e.g. constraining (controlling) the pathology variability significantly, there is no reason for the approach to be sufficiently sensitive in these comparably small samples. However, we conducted such an additional analysis (see response to a similar request R2.1). That being said, severe multicollinearity of activation-based CR scores and pathology could impact coefficient estimation (which was handled using cross-validation and bootstrapping) and interpretation. We have conducted additional correlation analyses of the CR score with different variables including pathological measures, suggesting minor associations ($r < 0.3$). Moreover, we performed a mediation analysis, which was added to the supplementary.

In the supplementary results we included the following section:

"In an attempt to understand why certain participants are able to maintain activation patterns, we investigated potential predictors and contributors of the CR score. In this sample, neither resting-state functional connectivity within seven standard networks nor mean global task-fMRI signal were significantly related to CR score variability (see Fig. S2). However, higher CR scores were associated with less pathological measures of AD burden such as the (squared) PL score ($p = 2.84 \cdot 10^{-5}$, $r = -0.272$ [-0.387,-0.148], $df = 229$), its components $A\beta_{42-40}$ ($p = 2.93 \cdot 10^{-4}$, $r = 0.189$ [0.110,0.354], $df = 229$), p -tau ($p = 0.017$, $r = -0.166$ [-0.289,-0.037], $df = 229$) and hippocampal volume ($p = 3.65 \cdot 10^{-6}$, $r = 0.208$ [0.121,0.291], $df = 487$), as well as lower global white matter hyper-intensity volumes ($p = 0.035$, $r = -0.098$ [-0.188,-0.007], $df = 460$). The CR score was further weakly positively correlated with total GM volumes in the regions with significant contributions to CR ($p = 0.013$, $r = 0.112$ [0.024,0.199], $df = 487$) and mean cortical thickness ($p = 0.023$, $r = 0.103$ [0.015,0.190], $df = 487$). Yet, it should be noted that accounting for these structural differences as covariates had essentially no effect on the observed results, neither in the multivariate moderation model nor the validation analyses of the CR score (see Fig. S19) and Tab. S1. Taken together, although AD pathology indices, tissue volumes and white matter lesions were slightly associated with individual CR score differences, the pattern of CR-related brain areas and its predictive value for memory performance was not mediated by atrophy alone or network connectivity at rest.

Due to the relationship between PL and the CR score, we also tested a formal mediation, which indicated that the CR score mediates the effect of PL (squared) on PACC5 (average causal mediation effect: -0.1566 [-0.3229, -0.0300], $p = 0.013$)."

We further address the mediation in the discussion:

"Besides the moderation effect, the CR score, which represents CR-related activity during successful memory encoding, was also weakly correlated with different measures of pathology and mediated the effect of PL on PACC5 scores (see supplementary). These findings point to a more intricate relationship between pathology, brain activity and cognitive performance, where low CR/hyperactivity and pathology could promote each in a vicious cycle^{31,42,43,44,45,46}."

Q3.4 It is interesting that the voxels contributing to the CR effect seem to be spared from atrophy. They may be contributing to CR by compensating for regions that are more impacted by pathology/atrophy. Perhaps this also presents an opportunity to address the previous review comment? Models that control for the extent of cortical atrophy would indicate that differences in brain activity are not simply reflecting a greater extent of neurodegeneration. This could be done with voxelwise maps as covariates, or even with a summary metric indicating the extent of cortical neurodegeneration.

R3.4 We thank the reviewer for bringing up this interesting issue. The analysis results do indeed suggest a weak association of structural measures such as mean cortical thickness (from Freesurfer; $r = 0.103$) or mean gray matter volume in the significant CR voxels (from SPM segmentation; $r = 0.112$) with our CR summary score, indicating a certain association of structural integrity with CR. However, neither the use of cortical thickness nor the GM volumes in the multivariate moderation model as additional covariate influenced the results significantly. For instance, the correlation of the voxelwise moderation (=CR) coefficients when including GM volumes as covariates versus not was 0.995 and 0.999 when covarying for mean cortical thickness. This suggests that the functional CR differences that we identified are not mediated by these morphological measures reflecting individual differences and atrophy. In addition, repeating our cross-sectional and longitudinal validation analyses with either of those two variables as a covariate did not affect the significance of the moderation effect by the CR score. The additional results are presented in the supplementary and read as follows.

"As seen in the previous section, there was a weak correlation of the CR score with structural measures, namely mean cortical thickness (obtained from Freesurfer) and mean gray matter volumes (from segmentation with SPM) in the voxels with significant contributions to CR. To ensure that the CR score represents reserve beyond mere structural integrity, we added a corresponding covariate to the multivariate moderation model. The results were found to be essentially the same as before, with correlations of 0.999 for mean cortical thickness and 0.995 for the mean GM volumes (see Fig. S19). Furthermore, we performed additional analyses analogous to the validation analyses in the main text, but with an additional morphometric covariate. Once again, the results were very similar, with only minor differences in the p values of the interaction effect of the CR score (see Tab. S1)."

Additionally, Table S1 was added, which includes the p values for the original model as compared to the models including the morphometric covariates. Figure S19 was also added, shown below:

Fig. S19: **CR coefficients when including morphometric covariates.** (A) Results of the multivariate model when including mean cortical thickness (from Freesurfer) as an additional covariate ($r = 0.999$ with CR coefficients in main text). (B) Results of the multivariate model when including mean GM volumes (from SPM segmentation) of the voxels with significant CR contributions (according to the main model) as an additional covariate ($r = 0.995$ with CR coefficients in the main text).

Q3.5 The cross-sectional analysis focuses on the moderation effect of the CR pattern, consistent with the definition of CR given in the Introduction. However, the longitudinal analysis does not include a similar moderation effect on pathology. There were other discrepancies between cross-sectional and longitudinal models that should also be resolved or explained, such as the inclusion of BAE and diagnostic group as covariates.

R3.5 We very much appreciate the helpful remark. Diagnostic group had been included to account for baseline differences in cognition in the whole sample instead of baseline PL, which was only available in about half of the participants due to a lack of CSF data in the other half. BAE was not included in this model as CR had entered the longitudinal model already in the form of a main effect. However, the reviewer's comment made us carefully reconsider our choices to better align cross-sectional and longitudinal analyses. We agree that the models should be more consistent, hence BAE is now included in the longitudinal model as is its interaction with time. Furthermore, in concordance with the consensus framework, we also included a three-way interaction between the CR score, pathology and time. However, due to the lack of PL scores in a large portion of the sample, we use squared hippocampal atrophy to (incompletely) represent pathology. While previously hippocampal volumes had been included in the cross-sectional model for the subsample without CSF data, these were now converted to hippocampal atrophy measures ranging from 0 to 1, thus being more consistent with the PL score. Both cross-sectionally and longitudinally, hippocampal "atrophy" is now used instead of hippocampal volumes. The longitudinal results for the subsample with PL scores are presented in the supplementary.

We revised parts of the results section, which now reads as follows:

"In a longitudinal context, lower hippocampal volumes at baseline worsened cognitive decline rates in the PACC5 score ($p = 5.24 \cdot 10^{-8}$, $\beta = -0.140 [-0.191, -0.092]$, $t = -5.717$, $df \approx 158.6$) in the whole sample. The CR score attenuated this relationship (three-way interaction of CR score, atrophy and time; $p = 1.19 \cdot 10^{-4}$, $\beta = 0.118 [0.060, 0.179]$, $t = 3.974$, $df \approx 124.5$), suggesting that activity patterns during memory encoding hold the potential to influence cognitive trajectories in the face of pathology (Fig. 6C and D). There was also an interaction of the CR score with hippocampal atrophy ($p = 3.28 \cdot 10^{-11}$, $\beta = 0.338 [0.240, 0.436]$, $t = 6.790$, $df \approx 487.6$) as in the cross-sectional models. Results for these analyses

when using the PL score instead of hippocampal atrophy in the CSF-subsample are presented in the supplementary.”

Furthermore, we adjusted the methods section accordingly:

"Hippocampal atrophy ranged from 0 to 1 like the PL score, with higher values representing more atrophy, and was obtained by multiplying the TIV-corrected hippocampal volumes with -1 and then re-scaling them. [...] Since the above model training and analyses were cross-sectional, as a final validation step, we utilized linear mixed-effects modeling (package lme4 in R) to test the moderation effect between pathology and the CR score longitudinally. The model included a subject-specific intercept and slope, as a model comparison had suggested a model with both random intercept and slope as superior compared to one with a random intercept alone. The model examined the three-way interaction effect between the CR score, hippocampal atrophy (squared) and time between measurements (continuous variable). It also included the corresponding two-way interactions. Hippocampal atrophy was used to maximize sample size for the longitudinal analysis. However, results for a similar model in the CSF-subsample using PL (squared) as pathology measure are presented in the supplementary. Age at baseline, sex, site of data acquisition, BAE and a BAE by time interaction were included as covariates."

The revised figure 6 is shown below:

Fig. 6: CR score is linked to cognitive performance cross-sectionally and longitudinally. (A) The relationship between the PL score and cognitive performance at baseline is moderated by the CR score. Cognitive performance is represented by three different scores: a global cognitive factor score, a memory factor score and

the PACC5 score (previously used for identification of the CR-related activity pattern). Cognitive performance was predicted using the respective regression model for an average individual with high (above median; red curve) or low (below median; blue curve) CR score. (B) In the sample without PL score, the CR score moderates the relationship between hippocampal atrophy and cognitive performance. (C) The pathology-dependent differences in longitudinal trajectories of cognitive performance are ameliorated by the baseline CR score. The PACC5 scores at a 5-year follow-up were predicted using the previously described LME (see methods section) fitted on the original longitudinal data for an average individual with high (above median; red line) or low (below median; blue line) CR score and high or low pathology (here represented by high and low hippocampal atrophy corresponding to the 25th and 75th percentile). (D) Predictions of all individual cognitive trajectories. Individuals were categorized as high/low CR based on their above/below average CR scores and as low/high pathology based on their below/above average hippocampal atrophy. Shaded areas in panels A-C denote 95% confidence intervals.

Q3.6 Minor revision/editing/language suggestions:

Q3.6a Lines 4-5 “tested this hypothesis in the Alzheimer’s disease continuum” – the sample description appears to include a majority of individuals that are not necessarily on the AD continuum unless there is evidence of abnormal biomarkers (152 CN, 51 participants with relatives with AD, and 202 participants with subjective cognitive decline).

R3.6a It is true that our sample primarily includes cognitively unimpaired individuals, of whom a large proportion is at risk for development for AD due to SCD and about half of the cognitively unimpaired individuals have abnormal AD biomarkers.

We have changed our wording from “*Alzheimer’s disease continuum*” to “*continuum from cognitively normal to at-risk stages for AD to AD dementia*”.

Q3.6b Line 17 “as the primary mechanism of CR” – one possible mechanism of CR was tested in this study, and results are unable to determine if it is the primary mechanism of CR.

R3.6b We thank the reviewer for the thoughtful comment and we have toned down our wording. We carefully rephrased the statement in the abstract to “*Our findings primarily provide evidence for the maintenance of core cognitive circuits including the DMN as a potential mechanism contributing to CR.*”

Additionally, we changed our wording in the discussion from “*In conclusion, rather than relying on the recruitment of additional brain regions as a compensatory mechanism, our findings point towards CR factors operative within core circuits themselves*” to “*In conclusion, our findings primarily support the notion that some CR factors might operate within core circuits themselves above compensatory activity discordant with successful encoding activity.*” We also included a sentence into the last paragraph of the discussion that now reads “*However, adequate judgment about compensation in the context of cognitive reserve should be based on further studies specifically designed for its investigation, involving multi-task and -contrast information as well as manipulation of task demand.*”

Q3.6c Lines 140-142: “which suggested the pivotal role of education in shaping how AD pathology influences cognitive abilities” implies causal associations based on a correlational analysis.

R3.6c We apologize for the misleading statement suggesting that we intended to make statements about causality. However, being aware of the limitation of the design and analysis methods used, we did not aim to imply that education directly influences the well-known relationship between AD pathology and cognitive performance outcomes. We carefully rephrased it to "*which suggested the pivotal role of education in promoting factors that might contribute to the relationship between AD pathology and cognitive abilities (Fig. 1B).*"

Q3.6d Lines 157-159, 170: "patterns of brain regions contributing both positively or negatively to the moderation of AD pathology" – may be more accurate to state 'to the moderation of the relationship between AD pathology and cognitive performance'

R3.6d We adjusted the lines accordingly thanks to the reviewer's suggestion.

Q3.6e Please clarify the following lines 212-215: "Taken together, the correlation between both patterns stood at 0.384, underlining that predominantly more of the typical i.e. activation/deactivation can support cognitive functioning while region-specific multifaceted relationships between these neural signatures and cognitive reserve might exist."

R3.6e We apologize for the lack of clarity or confusion this might have caused. We rephrased the sentence to "*Taken together, the correlation between the voxelwise CR coefficients and SM contrast values were found to be 0.384. This suggests that predominantly showing activation patterns closer to the typical activation/deactivation might support cognitive functioning. On the other hand, more complex region-specific multifaceted relationships between these neural signatures and cognitive reserve might exist, indicated e.g. by discordant voxels.*" and hope to have resolved the unclarity this way.

Q3.6f Lines 223-224: Instead of "high levels of brain activity", what about "higher SM contrast values? This would capture both higher and lower activity related to subsequent memory.

R3.6f We thank the reviewer for the suggestion, which we implemented.

Q3.6g It would be useful to provide some discussion around the alternate and statistically equivalent interpretation of the main moderation findings, i.e., rather than higher CR scores moderating the relationship between AD-related pathology and cognitive performance, higher pathology scores moderate the relationship between CR scores and cognitive performance.

R3.6g We thank the reviewer for the insightful comment. We agree that statistically speaking the inherent symmetries of the implemented moderation terms would also allow for an alternative interpretation in which higher pathology might moderate the relationship between CR scores and cognitive performance. However, considering the positive moderation coefficient, this would suggest that at the same levels of brain activity, higher pathological load would relate to better cognitive performance. From a biological standpoint this seems less plausible.

Moreover, we address this point in the revised limitation section. However, we would prefer to abstain from too much speculation.

“The approach was applied under the simple working assumption that pathology is the initial driver of cognitive decline, and that individual variations of task-related activity potentially affect this relationship. However, since the proposed main approach using cross-sectional data exploits simple correlations and symmetric interaction terms, it does allow for several alternative causal interpretations with inter-changed roles of key variables. Future longitudinal studies might focus deeper on the empirical plausibility of these alternative patterns of interplay.”

Q3.6h Please clarify what the p-values in lines 242-245 correspond to.

R3.6h We modified the sentence in order to improve clarity as follows:

“Importantly, this moderating effect was not only observed in individuals with cognitive impairment, i.e. aMCI and AD patients, but also when analyzing the same models only in cognitively unimpaired individuals (memory factor: $p = 3.91 \cdot 10^{-5}$, $\beta = 0.273$ [0.145,0.400], $t = 4.223$, $df = 171$; domain-general factor: $p = 0.0010$, $\beta = 0.220$ [0.091,0.349], $t = 3.358$, $df = 171$; PACC5: $p = 3.77 \cdot 10^{-6}$, $\beta = 0.301$ [0.177,0.426], $t = 4.781$, $df = 170$). This emphasizes that the fMRI activity patterns associated with CR might benefit a broad spectrum of cognitive abilities.”

In closing, we appreciate the reviewers' insightful comments and guidance throughout the revision process. We believe the amendments made have substantially enhanced the manuscript, addressing the key concerns effectively. We remain committed to refining our work as needed and look forward to any further suggestions that might improve our submission. Thank you for considering our revisions and for the opportunity to enhance our manuscript.

References:

Liu, T. T. (2016). Noise contributions to the fMRI signal: An overview. *NeuroImage*, *143*, 141-151.

Reviewer #1 (Remarks to the Author):

The authors are to be commended for a highly impressive rebuttal, matching the original paper in rigor and scope. The primary concerns raised in my prior review have been satisfied by the author's revisions, though one minor issue persists.

The authors refer to their findings several times throughout the manuscript (including the abstract) as a "mechanism" for cognitive reserve. Beyond the sense that causality is not established and the relationships remain associational, there is still the question of whether the authors are showing a "consequence" or "reflection" of reserve, rather than a mechanism. The authors should consider rewording the word "mechanism" throughout the manuscript.

I do not consider this to be a major impediment to the publication of this excellent work and, once the comment is satisfied, I would enthusiastically endorse the work for publication.

Reviewer #1 (Remarks on code availability):

I did not check to try to reproduce the code and I am not a matlab user, but the code seems appropriately documented and seems to cover the important analyses.

Reviewer #3 (Remarks to the Author):

The reviewers have done an excellent job in responding to previous comments and have addressed my primary concerns. I just have one clarification to R3.6g regarding the alternative interpretation of the interaction effect. I leave it up to the authors whether this is included in the paper, but in my mind it supports the interpretation of the CR score as reflecting resilience.

The interaction is framed as higher CR scores weakening the association of pathology on cognition. We pointed out that the interaction has an alternative interpretation (i.e., flipping which variable is considered the moderator) In the response, the authors wrote that the alternative interpretation would be "at the same levels of brain activity, higher pathological load would relate to better cognitive performance." I think this would be accurate when including just the main effect of brain activity. Regarding the interaction, I would instead propose the alternative interpretation that, at higher levels of pathology, the association between CR and cognitive performance is stronger. Conversely, at low levels of pathology, the association of CR and cognitive performance is weaker. This is consistent with a conceptualization of resilience in which it only exerts a detectable effect in the presence of risk. Figure 6C is a nice demonstration of this.

Response Letter: NCOMMS-23-49941B

We thank the editor and reviewers again for their in-depth constructive comments and helpful suggestions that helped strengthening the manuscript. We addressed the remaining comments.

REVIEWERS' COMMENTS

Reviewer #1 (Remarks to the Author):

Q1.0 The authors are to be commended for a highly impressive rebuttal, matching the original paper in rigor and scope. The primary concerns raised in my prior review have been satisfied by the author's revisions, though one minor issue persists.

R1.0 We thank the reviewer again for the helpful comments and the repeated positive evaluation of our work.

Q1.1 The authors refer to their findings several times throughout the manuscript (including the abstract) as a "mechanism" for cognitive reserve. Beyond the sense that causality is not established and the relationships remain associational, there is still the question of whether the authors are showing a "consequence" or "reflection" of reserve, rather than a mechanism. The authors should consider rewording the word "mechanism" throughout the manuscript. I do not consider this to be a major impediment to the publication of this excellent work and, once the comment is satisfied, I would enthusiastically endorse the work for publication.

R1.1 We thank the reviewer for the accurate comment. We agree that the current work is not enough to establish definitive causality, even though our longitudinal results provide some support for such a hypothesis. We toned down the wording and changed the sentence in the abstract from "*Our findings primarily provide evidence for the maintenance of core cognitive circuits including the DMN as the potential mechanism contributing to CR.*" to "*Our findings primarily provide evidence for the maintenance of core cognitive circuits including the DMN as the neural basis of CR.*" We further exchanged the word "mechanism" by "implementation" in the discussion, where the sentence now reads "*Overall, our findings suggest an enhanced maintenance of core cognitive circuits as the primary neural implementation of cognitive reserve.*"

Reviewer #3 (Remarks to the Author):

Q3.0 The reviewers have done an excellent job in responding to previous comments and have addressed my primary concerns. I just have one clarification to R3.6g regarding the alternative interpretation of the interaction effect. I leave it up to the authors whether this is included in the paper, but in my mind it supports the interpretation of the CR score as reflecting resilience.

R3.0 We thank the reviewer for this positive judgement.

Q3.1 The interaction is framed as higher CR scores weakening the association of pathology on cognition. We pointed out that the interaction has an alternative interpretation (i.e., flipping which variable is considered the moderator) In the response, the authors wrote that the alternative interpretation would be “at the same levels of brain activity, higher pathological load would relate to better cognitive performance.” I think this would be accurate when including just the main effect of brain activity. Regarding the interaction, I would instead propose the alternative interpretation that, at higher levels of pathology, the association between CR and cognitive performance is stronger. Conversely, at low levels of pathology, the association of CR and cognitive performance is weaker. This is consistent with a conceptualization of resilience in which it only exerts a detectable effect in the presence of risk. Figure 6C is a nice demonstration of this.

R3.1 We thank the reviewer for the clarification and found it noteworthy to point out the analogous interpretation. Hence, we included the sentence “*Analogously, individuals with higher pathological burden benefit more from higher levels of CR compared to the ones with low pathology.*” in the discussion.